# Improved estimates of smoke exposure during Australia fire seasons: Importance of quantifying plume injection heights

Xu Feng[1], Loretta J. Mickley[1], Michelle L. Bell[2], Tianjia Liu[3], Jenny A. Fisher[4], Maria Val Martin[5]

[1] John A. Paulson School of Engineering and Applied Sciences, Harvard University, Cambridge, MA, 02138, USA
[2] School of Environment, Yale University, New Haven, CT, 06511, USA
[3] Department of Earth System Science, University of California, Irvine, CA, 92697, USA
[4] Centre for Atmospheric Chemistry, School of Earth, Atmospheric and Life Sciences, University of Wollongong, Wollongong, 2522, Australia
[5] School of Biosciences, University of Sheffield, Sheffield, S10 2TN, UK

*Correspondence to*: Xu Feng (xfeng@g.harvard.edu)

**Abstract.**

Wildfires can have a significant impact on air quality in Australia during severe burning seasons, but
incomplete knowledge of the injection heights of smoke plumes poses a challenge for quantifying smoke exposure. In this study, we use two approaches to quantify the fractions of fire emissions injected above the planetary boundary layer (PBL), and we further investigate the impact of plume injection fractions on daily mean surface concentrations of fine particulate matter ($PM_{2.5}$) from wildfire smoke in key cities over northern and southeastern Australia from 2009 to 2020. For the first method, we rely on climatological,
monthly mean vertical profiles of smoke emissions from the Integrated Monitoring and Modelling System for wildland fires (IS4FIRES), together with assimilated PBL heights from NASA Modern-Era Retrospective analysis for Research and Application (MERRA) version 2. For the second method, we develop a novel approach based on the Multi-angle Imaging Spectro-Radiometer (MISR) observations and a random forest, machine-learning model that allows us to directly predict the daily plume injection
fractions above the PBL in each grid cell. We apply the resulting plume injection fractions quantified by the two methods to smoke $PM_{2.5}$ concentrations simulated by the Stochastic Time-Inverted Lagrangian Transport (STILT) model in target cities. We find that characterization of the plume injection heights greatly affects estimates of surface daily smoke $PM_{2.5}$, especially during severe wildfire seasons, when intense heat from fires can loft smoke high in the troposphere. However, using climatological injection
profiles cannot capture well the spatiotemporal variability of plume injection fractions, resulting in a 63% underestimate of daily fire emission fluxes injected above the PBL in comparison with those fluxes derived from MISR injection fractions. Our random forest model successfully reproduces the daily injected fire emission fluxes against MISR observations ($R^2 = 0.88$, normalized mean bias = 10%), which predicts that 27% and 45% of total fire emissions rise above the PBL in northern and southeastern Australia, respectively,

from 2009 to 2020. Using the plume behavior predicted by the random forest method also leads to better model agreement with observed surface $PM_{2.5}$ in several key cities near the wildfire source regions, with smoke $PM_{2.5}$ accounting for 5% to 52% of total $PM_{2.5}$ during fire seasons from 2009 to 2020.

# 1 Introduction

Wildfire is a strong seasonal source of air pollution in Australia, significantly contributing to poor air quality especially during severe burning seasons such as the "Black Summer" in 2019 (e.g., Reisen et al., 2011; Aryal et al., 2018; Ryan et al., 2021; Graham et al., 2021). The peak periods of wildfires over northern Australia are generally during the dry season (April to October), when the high-pressure systems located in southern Australia bring dry and warm southeasterly winds to the Top End and Far North Queensland

(FNQ) (Australian Bureau of Meteorology, 2023b). The Australian monsoon also governs fire seasonality in northern Australia. During the monsoon periods from November to April, the prevailing winds shift to northwesterly, bringing moist air from the ocean and reducing the risk of wildfires (Australian Bureau of Meteorology, 2023a). In southern Australia, the burning season typically occurs in austral spring and summer (September to February) when fuels are abundant. However, fire activity in this region shows large

interannual variability. The fire danger increases when low-pressure systems in Tasmania bring hot and dry westerly winds to the coastal areas (Australian Bureau of Meteorology, 2023b).

Smoke emitted from wildfires is a complex mixture of organic carbon (OC), black carbon (BC), and other types of fine particulate matter ($PM_{2.5}$), together with a suite of trace gases. Smoke $PM_{2.5}$ is harmful to human health and the ambient environment (Reid et al., 2016; Aguilera et al., 2021; Johnston et al., 2021).

There are large uncertainties, however, in estimates of exposure to smoke $PM_{2.5}$ downwind of fires, in part because the transport of wildfire plumes is challenging to quantify in space and time. In Australia, most fire emissions are released in the planetary boundary layer (PBL), but sufficient buoyancy generated by the heat from intense wildfires can inject emissions into the free troposphere or even stratosphere (Fromm et al., 2006; Dirksen et al., 2009; Mims et al., 2010; Val Martin et al., 2018; Solomon et al., 2022). Val Martin et

al. (2018) showed that significant fractions (5% to 25%) of total column biomass burning emissions were injected above 2 km in Australia during the summer months from 2008 to 2010. The plume injection heights determine the vertical distribution of fire emissions, affecting surface smoke exposure, the long-range transport, and removal processes of emitted pollutants (e.g., Jian and Fu, 2014; Zhu et al., 2018). A recent study used three plume rise schemes in the Community Multiscale Air Quality model to study the plume

injection heights and their impacts on air quality, indicating that higher plume injection heights led to lower surface $PM_{2.5}$ concentrations near the source region but higher concentrations in regions downwind due to the transport at higher altitudes followed by downward mixing (Li et al., 2023). Here, we develop two methods to quantify the fraction of fire emissions injected above the PBL, and further investigate the impacts of plume injection heights on daily smoke $PM_{2.5}$ over Australia.

Previous studies have retrieved the plume injection heights and estimated the climatological injection profiles from satellite data, including from the Multi-angle Imaging Spectro-Radiometer (MISR), the

Cloud-Aerosol Lidar with Orthogonal Polarization (CALIOP) instruments (Kahn et al., 2007; Tosca et al., 2011; Raffuse et al., 2012; Paugam et al., 2016; Val Martin et al., 2010; 2018), and the TROPOspheric Monitoring Instrument (TROPOMI, Griffin et al., 2020). These approaches have drawbacks. For example, MISR and CALIOP provide global coverage every nine days and every sixteen days, respectively, near the equator, though more frequently at high latitudes. These instruments thus may miss fire occurrences due to their inadequate temporal resolution and the narrow detection swath. In addition, digitizing the plumes of MISR imagery is both labor intensive and computationally expensive, resulting in limited datasets of plume injection heights (Nelson et al., 2013; Val Martin et al., 2018). The plume heights retrieved from TROPOMI offer daily global coverage, but TROPOMI data are available only from 2018 onwards and so cannot be utilized for a long-term study.

To address these issues, several biomass burning emission inventories have incorporated information on injection height at high spatiotemporal resolution. These include the Global Fire Assimilation System (GFAS, Rémy et al., 2017) and the Integrated Monitoring and Modelling System for Wildland Fires (IS4FIRES, Sofiev et al., 2009; Soares et al., 2015). Both GFAS and IS4FIRES rely on a plume rise model (Freitas et al., 2007, 2010) and semi-empirical parameterization (Sofiev et al., 2012; 2013) to determine injection heights. Besides these two methods for estimating injection height, Yao et al. (2018) used a machine learning model (random forest) and CALIOP data to predict the minimum heights of forest fire smoke in Canada. These three datasets represent the vertical extent of smoke plumes with high-resolution single parameters that specified the top and bottom heights of plumes, as well as the mean height of maximum injection (MHMI). The definitions of these variables are described in appendix S1. However, such parameters do not quantify the fraction of fire emissions within the PBL, a critical value for quantifying smoke exposure within the PBL. IS4FIRES also provides climatological, monthly mean profiles of plume injection heights, which do specify the fire emissions that remain within the PBL. But this climatological dataset cannot capture the large interannual variability of plume injection heights and wildfire intensity (Val Martin et al., 2010; 2018).

Another challenge in calculating smoke exposure involves the modeling of smoke plume transport. Previous studies have applied multiple modeling techniques to capture transport, including use of 3-D offline or online coupled atmospheric chemistry models (e.g., Fann et al., 2018; Liu et al., 2017; Gan et al., 2017) and Lagrangian particle dispersion models such as HYSPLIT or STILT (e.g., Thelen et al., 2013; Mallia et al., 2015). The 3-D chemistry models can simulate the physical and chemical processes of smoke $PM_{2.5}$ based on the biomass burning emission inventory but are computationally expensive for long-term simulations at fine spatial resolution (Johnson et al., 2020). In contrast, Lagrangian modeling applies a receptor-oriented framework, allowing (1) computationally efficient tracking of the smoke emitted across a finely gridded, large spatial domain and (2) determination of the contributions of smoke to the air quality

in the receptor city downwind. This modeling framework performs better in terms of numerical stability and mass conservation than do 3-D models (Lin et al., 2013; Wohltmann and Rex, 2009). However, Lagranigian modeling usually lacks chemical processes and is unable to capture background $PM_{2.5}$ concentrations from other anthropogenic and natural sources.

Many studies on wildfire smoke exposure in Australia are based on ground-based observations (e.g., Morgan et al., 2010; Johnston et al., 2021; Cortes-Ramirez et al., 2022). These studies usually use statistical methods to separate the smoke $PM_{2.5}$ from background $PM_{2.5}$, as the air quality monitors measure only total $PM_{2.5}$. This method, however, is unable to determine the spatial distribution of smoke emissions that contribute to the observed $PM_{2.5}$. Alternatively, some studies use atmospheric chemistry models to
explicitly simulate smoke $PM_{2.5}$ concentrations from open fires and their impacts on air quality and health in Australia (Rea et al., 2016; Nguyen et al., 2020, 2021; Graham et al., 2021). These studies can provide more accurate spatiotemporal variability of smoke air quality but may focus only on short-term simulations due to computational expense. Furthermore, the accuracy of simulated smoke $PM_{2.5}$ concentrations in these models depends on reliable meteorology, biomass burning emissions, and plume injection heights.

In this paper, we build on past efforts to model smoke exposure in Australia. Our goal is to improve the accuracy of smoke exposure in the receptor cities by better quantifying the fraction of smoke plumes remaining in the PBL across northern and southeastern Australia. We also quantify the source regions of smoke $PM_{2.5}$ in these cities. We first focus on two methods to quantify the daily fractions of fire emissions within and above the PBL: (1) the climatological injection profiles from IS4FIRES and (2) plume injection
heights from MISR observations. Both methods are described in Section 2. We apply the predicted injection fractions to the Lagrangian plume model STILT to simulate the daily smoke $PM_{2.5}$ in key cities across Australia during the fire seasons from 2009 to 2020. In Sections 3 and 4, we compare the plume injection fractions predicted by our two methods, and we validate the derived smoke $PM_{2.5}$ concentrations against the surface $PM_{2.5}$ observations.

**2 Methods and data**

**2.1 Estimation of plume injection fractions using climatological injection profiles**

     We estimate the fractions of smoke plumes injected above the PBL using two methods. In the first method, we first compare the daily PBL and the plume injection heights for each fire event. For those plumes that rise above the PBL, we use the climatological, monthly mean profile of plume injection heights in that grid
cell to apportion smoke abundance within the PBL and above it.

Daily mean PBL heights across Australia are obtained from the Modern-Era Retrospective Analysis for Research and Applications Version 2 (MERRA-2, Gelaro et al., 2017) at a spatial resolution of 0.5° latitude × 0.625° longitude. This reanalysis is often used to drive chemical transport models such as GEOS-Chem (Bey et al. 2001; Keller et al., 2014; Kim et al., 2015). We use the daily injection heights compiled by the

GFAS emission inventory (Rémy et al., 2017), which provides four parameters representing the vertical extent of each smoke plume at 0.1° × 0.1° resolution: the top and bottom heights of plumes, the MHMI, and injection height. These parameters are calculated with two distinct algorithms: the one-dimensional plume rise model (Freitas et al., 2007, 2010; Rémy et al., 2017) and the IS4FIRES parameterization (Sofiev et al., 2012; 2013). The plume rise model predicts the daily vertical velocity, horizontal plume velocity,

temperature, and plume radius; the model relies on assimilated meteorology from the European Centre for Medium-Range Weather Forecasts (ECMWF) and active fire area retrieved from the Moderate Resolution Imaging Spectroradiometer (MODIS). In contrast, IS4FIRES calculates the daily plume injection height based on fire radiative power (FRP) from MODIS as well as on ECMWF meteorology (Sofiev et al., 2012).

In addition to plume height, we also determine the mass fraction of smoke emitted above the PBL.

IS4FIRES also offers global maps of monthly mean injection profiles of fire emissions at a spatial resolution of 1° × 1° × 500 m from the surface to 10 km altitude (20 layers), averaged over the years 2000 to 2012 (http://is4fires.fmi.fi, last accessed: October 21, 2022). The IS4FIRES parameterization assumes that each fire lasts for 24 hours and that the plume heights of this fire depend on fire intensity, which is based on the mean diurnal variation of the FRP derived from the geostationary orbiting instrument Spinning Enhanced

Visible and Infrared Imager (Roberts et al., 2009, Sofiev et al., 2013). The resulting hourly injection profiles are averaged over the whole day and aggregated to the monthly level. The profiles are then normalized by monthly mean emissions in that vertical column. More details are described in Sofiev et al. (2013).

In this study, we match the plume top heights from GFAS and the meteorology from MERRA-2 values to align with 0.25° × 0.25° resolution of the Global Fire Emission Database Version 4.1 inventory, (GFED

4.1s, van der Werf et al., 2017). Since regridding the GFAS plume top heights would lead to excessive smoothing, we choose the largest height within each GFED grid cell, matching plume by plume. In some cases, we needed to search a 1° border around the GFED grid cell to locate the corresponding plume in GFAS. We follow a similar approach for matching MERRA-2 meteorology with the GFAS plume top heights, choosing the largest PBL height in the GFED grid cell. We then compare the MHMI derived from

the plume rise model with the associated daily mean PBL height from MERRA-2 to determine whether the fire emission should be lifted above the PBL at each grid cell. We assume that total fire emissions remain within the PBL if the daily mean PBL height ($H_{pbl}$) is higher than the MHMI ($H_{MHMI}$). For those grid cells in which the MHMI is higher than the PBL heights, we calculate the daily injection fractions of fire emissions above the PBL as follows:

$$
\qquad f_{abovePBL}(i,j,d) = \begin{cases} 0, & H_{MHMI}(i,j,d) < H_{pbl}(i,j,d) \\ \displaystyle\sum_{k=Z_{pbl}}^{Z_{top}} f(i,j,k,m), & H_{MHMI}(i,j,d) \geq H_{pbl}(i,j,d) \end{cases} \qquad (1)
$$

where $f_{abovePBL}(i,j,d)$ is the daily injection fractions at location $(i,j)$ over the day $d$, and $f(i,j,k,m)$ is the monthly mean normalized vertical fraction of fire emissions injected into the layer $k$ in month $m$, calculated by the IS4FIRES parameterization. We sum up the fractional fire emissions $f(i,j,k,m)$ from $Z_{pbl}$, the vertical layer where the daily mean PBL height $H_{pbl}(i,j,d)$ is located, to the top layer of the normalized injection profile ($Z_{top}$). This yields the plume mass fraction above the PBL.

### 2.2 Estimation of plume injection fractions using machine learning models

#### 2.2.1 MISR data and target variable

We also develop a novel approach using random forest models to predict the fractions of smoke plumes injected above the PBL in each grid cell. The explanatory variables consist of satellite retrievals of plume heights, fire information, land use classification, and meteorological variables.

The plume heights used for training are those observed by the MISR instrument. MISR is on board Terra, a polar-orbiting satellite, overpassing the equator in the descending mode at 10:30 local time. MISR acquires imagery in four spectral bands along the orbiting track, using nine cameras with viewing angles from $\pm\,70.5°$ to $\pm\,26.1°$ relative to nadir. The four spectral bands are centered at wavelengths of 446 nm, 558 nm, 672 nm, and 866 nm (Diner et al., 1998). The swath width of MISR is 380 km, covering Australia every four to five days. Data acquired from the blue (446 nm) and red (672 nm) bands can be used to retrieve smoke plume heights at horizontal spatial resolutions of 1.1 km and 275 m, respectively. Although the red-band data have higher spatial resolution, the retrieval quality of the red band is usually worse than that of the blue band, especially for thin plumes over a bright surface such as is typical for Australia (Nelson et al., 2013).

The MISR Interactive eXplorer (MINX, https://github.com/nasa/MINX, last accessed: October 21, 2022) is an interactive software that digitizes the plume heights from MISR data, using a stereoscopic height retrieval algorithm (Nelson et al., 2008, 2013). Digitizing the plume heights using MINX is time-consuming as the perimeters of individual plumes need to be identified manually by users (described in Appendix S2). As a consequence, archived MISR retrievals of global plume heights are available only for a limited number of months – the years 2008 to 2011 and for June, July, and August of 2017 and 2018. These plume heights were calculated for the MISR Plume Height Project 2 (MPHP2, https://misr.jpl.nasa.gov/get-data/misr-plume-height-project-2/, last accessed: October 21, 2022).

For training and validating the random forest models, we collected 2212 records of plume height retrievals in Australia, including 2021 records from MPHP2 and 191 supplemental records that we generated using MINX for November 2019 during the severe wildfire season. These MISR plume records are mainly distributed over the coastal areas of northern and southern Australia (Figure S1). In general, each record represents one plume, but sometimes several plumes overlap. There may exist more than one record per plume, or one record may describe more than one plume. For each identified plume, MINX digitizes two retrievals of plume heights based on the blue-band and red-band data within the plume perimeter, each of which is classified as having "Good," "Fair," or "Poor" retrieval quality. We exclude plume records labeled "Poor." For all other plumes, we choose one record from either the blue-band or red-band data, depending on which band exhibits better retrieval quality. Here we use the zero-wind heights (described in Appendix S2) to calculate the vertical profile for each plume. We remove unrealistic heights lower than the terrain heights (i.e., when zero-wind height minus terrain height < 0 km), as well as those higher than 8 km above the local terrain. Negative zero-wind heights are due to the retrieval biases of pixels near to or on the ground, while heights greater than 8 km are likely an artefact caused by pyro-cumulus clouds overlaying the plumes (Val Martin et al., 2010). We obtain the injection profile by normalizing the vertical distribution of retrieved plume heights above local terrain in increments of 0.25 km altitude from 0 to 8 km for each plume. We then compute the injection fractions above the PBL based on Eq. (1), where the daily mean PBL height is the same as the data described in Section 2.1.

### 2.2.2 Predictors for random forest model

We use daily meteorological variables, fire information, and land use classifications as predictors (Table 1) for the random forest models. The meteorological variables are from MERRA-2 at 0.5° latitude × 0.625° longitude resolution and include the daily means of PBL height, air temperature at 2 m above the surface, surface relative humidity, U/V-wind at 10 m, and total precipitation. Fire information consists of the fire location for each plume and FRP, both from the MODIS/Terra Thermal Anomalies/Fire 5-Min L2 Swath 1km V061 (MOD14, Giglio and Justice, 2021). The MINX software calculates the total FRP of the digitized plume from this dataset. The daily fire emissions of OC from the GFED 4.1s are also incorporated into the random forest model. We use only OC emissions because variations in OC and BC, the other main component of smoke $PM_{2.5}$, are assumed to correlate. We sample the MERRA-2 grid and GFED grid closest to the initial source point of the smoke plume specified, based on MOD14. We also include the yearly land cover classification of the International Geosphere-Biosphere Programme (IGBP) derived from the MODIS Land Cover Climate Modeling Grid Version 6 (MCD12C1, Friedl and Sulla-Menashe, 2015) at 0.05° × 0.05° spatial resolution. Wildfires occurring in various vegetation types such as forest, shrubland, and grassland usually lead to different plume injection heights, which can be classified by land use data. The

MINX software diagnoses the land use type at the location with the maximum FRP within the digitized plume boundary.

### 2.2.3 Random forest algorithm

Random forest is a widely used machine learning method for both classification and regression, containing an ensemble of bootstrap aggregated, or "bagged," decision trees. Each individual decision tree is trained using a random sample of the training dataset to reduce the correlation between different decision trees. The final predictions of a random forest model are based on the average of predictions from each decision tree (Breiman, 2001). A decision tree is built by splitting the data into left and right nodes

recursively, based on the standard Classification And Regression Tree (CART) algorithm (Breiman, 2001). In node $p$, the mean squared error (MSE) is calculated as Eq. (2):

$$MSE(p) = \sum_{j \in P} \frac{1}{n}\left(y_j - \bar{y}_p\right)^2 \tag{2}$$

where $y_j$ and $\bar{y}_p$ are the target variable with observation index $j$ and the mean value of target variable samples in node $p$, respectively. $P$ represents the set of all observation indices in node $p$ and $n$ is the sample

size. The algorithm sorts one of the predictors $x_i$ ($i = 1,2,\dots,11$) and uses each element of $x_i$ as a split point to divide the samples into two subsets. The algorithm then calculates the decline in MSE ($\Delta MSE$) for each splitting point as Eq. (3):

$$\Delta MSE = \sum_{j \in P} \frac{1}{n}MSE(p) - \sum_{j \in P_L} \frac{1}{n}MSE(p_L) - \sum_{j \in P_R} \frac{1}{n}MSE(p_R) \tag{3}$$

where the $p_L$ and $p_R$ are the left and right nodes. The best split point is determined by maximizing the

decline in MSE ($\Delta MSE$). Each node will stop splitting when there are less than five samples within this node, which avoids overfitting on the training datasets. To estimate the importance of each predictor, the algorithm randomly permutes the values of each predictor within the dataset and calculates the increases in MSE over each decision tree, compared to the original set of MSEs. More important predictors will generate greater increases in MSE when permuted. The importance of each predictor is then indicated by its mean

value divided by the standard deviation of the increases in MSE over all decision trees.

In this study, we construct the random forest model with 100 regression decision trees. As noted above, Table 2 shows the predictors and the target variable (i.e., daily plume injection fractions above the PBL). Total records of the target variable and associated predictors are divided into a training dataset (n = 2012 records) and a test dataset (n = 200 records). We select as test data one record of every ten records in order

of observed dates, which ensures evenly sampling the whole dataset. We first train the random forest model using the training dataset and then apply the predictors from the test dataset to the resulting random forest

model. Validation is carried out by comparing the predictions with the true values of the target variable from the test dataset.

A shortcoming of our machine-learning approach is that the MISR dataset used for our study includes relatively few plumes in southeastern Australia compared to northern Australia (Figure S1). The fire season is shorter in this region, and there is much greater interannual variability in fire activity. As a consequence, we have available only 244 training records in the southeast and only 10 for testing there, compared to xxx records for training and xxx records for testing in the north. We further discuss this limitation in Section 5.

**2.3 Calculation of smoke PM$_{2.5}$ concentrations using the STILT model**

**2.3.1 STILT and fire emission inventory**

We use the STILT model version 2 (Lin et al., 2003, Fasoli et al., 2018) to simulate the daily smoke PM$_{2.5}$ concentrations in 12 key cities (shown in Table S1) over Australia during the fire seasons from the years 2009 to 2020. STILT is a Lagrangian particle dispersion model driven by assimilated meteorology from the National Oceanic and Atmospheric Administration Air Resources Laboratory and National Centers for Environmental Prediction (Stein et al., 2015). The model calculates "sensitivity footprints" in units of concentration divided by emissions (ppm μmol$^{-1}$ m$^2$ s), as described in appendix S3. These footprints relate potential emissions across a source region upwind of a given receptor to air pollution within the PBL at that receptor. As we describe below, multiplication of these footprints by emissions within the source region yields the concentration change in an air pollutant at the receptor. The model yields the concentrations of fire-related black carbon (BC) and organic carbon (OC) particulate matter at each receptor within the source region via multiplying the sensitivity footprints by the fire emissions on daily timescales. Smoke PM$_{2.5}$ is typically defined as the sum of the fire-related BC and organic matter (OM) (Chow et al., 2011; Koplitz et al., 2016; Cusworth et al., 2018; Li et al., 2020). OM is calculated using a mass ratio of OM to OC, which is assumed to be 2.1 (Philip et al., 2014).

We apply the fire emissions of OC and BC over Australia from the GFED 4.1s inventory (van der Werf et al., 2017), which compares well with other inventories for Australia (Liu et al., 2020; Desservettaz et al., 2022) and includes methodologies specifically designed to better capture small fires (Randerson et al., 2012). GFED 4.1s estimates the monthly emissions at 0.25° spatial resolution from 1997 to present based on the burned area data from MODIS MCD64A1 (Giglio et al., 2013). The monthly emissions are redistributed into daily timescales using daily scale factors determined by the MODIS active fire products (MCD14ML) and the burning day reported in MCD64A1 (van der Werf et al., 2017).

### 2.3.2 Setup of sensitivity experiments

We conduct three sensitivity experiments to evaluate the effects of plume injection fractions on the calculations of smoke $PM_{2.5}$ concentrations. Table 2 shows the configurations of the STILT model and the sensitivity experiments. The domain covers mainland Australia at $0.25° \times 0.25°$ spatial resolution, consistent with that of the GFED 4.1s inventory. The STILT simulations are driven by archived meteorological variables from the Global Data Assimilation System (GDAS) at $0.5° \times 0.5°$ resolution for 2009 to 2018 and from the Global Forecast System (GFS) at $0.25° \times 0.25°$ resolution for 2019 to 2020. STILT simulates the sensitivity footprints backwards in time for 120 hours, which allows the air parcels to travel the equivalent of the whole of Australia.

For the control experiment (Case CTL), we assume that all fire emissions are released within the PBL, where they are evenly distributed. Daily smoke $PM_{2.5}$ concentrations at the receptors are then derived from the total fire emissions of OM (scaled from OC) and BC multiplied by the simulated sensitivity footprints. For the two sensitivity experiments, we consider the impacts of plume injection on the surface concentrations of smoke $PM_{2.5}$ downwind. In both these cases, we scale the fire emissions by the fractions of smoke mass remaining within the PBL. We assume that the fire emissions injected above the PBL have no impact on the surface $PM_{2.5}$. For case INJ-CLIM, we estimate these fractions using climatological plume profiles (Section 2.1), and for case INJ-RF, we make these estimates using the random forest algorithm (Section 2.2). However, the INJ-CLIM and INJ-RF methods estimate the plume injection fractions only in the source grids, and they are unable to estimate to what extent smoke plumes mix down to the surface in remote regions downwind. This assumption may lead to the low biases of surface smoke $PM_{2.5}$ in remote regions, which we discuss in Section 4.

### 2.4 Calculation of non-fire $PM_{2.5}$ concentrations

To validate the simulated smoke $PM_{2.5}$, we need to estimate the contribution of non-fire $PM_{2.5}$ to total $PM_{2.5}$, as only measurements of total $PM_{2.5}$ are available (Section 2.5). To that end, we utilize the surface measurements of $PM_{2.5}$ on low-fire days (defined below) to calculate a non-fire $PM_{2.5}$ concentration for each year, as in Cusworth et al. (2018). For each receptor in a given year, we first define an upwind burning region as those grid cells where the mean simulated footprint sensitivities during the fire season are higher than a certain threshold, which we arbitrarily specify as $10^{-4}$ ppm $\mu mol^{-1}$ $m^2$ s. We then analyze the time series of daily OC fire emissions from the GFEDv4s inventory summed over all grid cells in this upwind burning region during the wildfire season every year and specify the $20^{th}$ percentile at the low end of the fire emissions frequency distribution as an emission threshold. We tag a day as "low-fire" if the daily OC fire emissions over the upwind burning region during the previous two days fall below the emission

threshold (Cusworth et al., 2018). The average of all $PM_{2.5}$ surface observations at the receptor during the
low-fire days is assumed to be the non-fire $PM_{2.5}$ concentration for the fire season in that year.

**2.5 Ground-based observations of $PM_{2.5}$ in Australia**

We rely on ground-based measurements of total $PM_{2.5}$ concentrations to validate the modeled smoke
$PM_{2.5}$. Table 3 shows the sites and time periods of the historical data used for this validation. These data
include hourly ground-based $PM_{2.5}$ observations from the Northern Territory Environment Protection
Authority (http://ntepa.webhop.net/NTEPA/Default.ltr.aspx, last accessed: June 7, 2023), the Victoria
Environment Protection Authority (https://www.epa.vic.gov.au/for-community/airwatch, last accessed:
October 21, 2022), the Queensland Government Open Data Portal (https://apps.des.qld.gov.au/air-
quality/download/, last accessed: June 7, 2023), the New South Wales Department of Planning and
Environment    (https://www.dpie.nsw.gov.au/air-quality/air-quality-data-services/data-download-facility,
last accessed: June 7, 2023), and the Australian Capital Territory Government Open Data Portal
(https://www.data.act.gov.au/Environment/Air-Quality-Monitoring-Data/94a5-zqnn, last accessed: June 7,
2023). Daily $PM_{2.5}$ concentrations are calculated as the average of the available hourly observations on each
day. We exclude the daily mean observations when more than eight values of the hourly data are missing
for that day.

**3 Plume injection fractions during Australian fire seasons**

**3.1 Wildfire activity in Australia**

Figure 1 shows the spatial distributions of annual mean total OC fire emissions averaged from 2009 to
2020, indicating that the northern and southeastern areas are the most fire-prone in Australia. In this study,
we focus on the regional smoke exposure in northern Australia (118.125°E-150.875°E, 18.875°S-10.125°S)
and southeastern Australia (140.125°E-153.875°E, 43.875°S-24.125°S, dashed boxes in Figure 1), where
seasonal wildfires produce 39.5% and 41.1% of total fire emissions in Australia, and where 80% of the
Australian population lives (Australian Bureau of Statistics, 2022). In northern Australia, the two main
burning regions are located in the Top End and FNQ, which are covered by eucalypt forests and woodlands.
In southeastern Australia, burning regions are mainly distributed in coastal eucalypt forested areas in New
South Wales and Victoria, as well as in the Australian Capital Territory. In this study, we focus on the
smoke exposure during April to December in northern Australia and August to January of the next year in
southeastern Australia. In 2020, fire activity in southeastern Australia continued to some extent into
February, but this lengthening of the typical fire season was unusual (Ellis et al., 2022).

### 3.2 Evaluation of plume injection fractions calculated by climatological injection profiles and predicted by random forest

Figure 2a compares the plume injection fractions above the PBL ($f_{abovePBL}$) derived from the MISR plume records with those calculated using the climatological plume profiles with assimilated PBL data (first method described in Section 2.1). There are 2212 samples in total. Each sample represents an individual plume digitized from the MISR imagery. Results show that the estimated daily plume injection fractions are inconsistent with MISR observations with a low correlation coefficient of 0.24 and a large root mean square error (RMSE) of 0.39, indicating that climatological profiles cannot reproduce the daily variation of plume injection fractions. We find that 90% of the overestimated injection fractions with relative low fire emissions are located in the north and central of Australia, a finding which we attribute to inaccuracies in the climatological plume profiles. The plume injection height of the plume profile is proportional to the PBL height in this method (appendix S1, Sofiev et al., 2013), and given the relatively deep PBL in this region, the injection fractions above the PBL tend to be overestimated. Next, we compare the observed and modeled fire emission fluxes in the atmosphere above the PBL (Figure 2b). These fluxes are calculated by scaling total emission fluxes from GFED 4.1s using injection fractions derived from the first method and from MISR observations (Eq. 1). We find that the climatological method can explain 76% of the variance in the injected emission fluxes derived from MISR, but still underestimates the mean value by 63%. The large bias is mainly due to the underestimates of injection fractions for some megafires, such as those in 2019. The intense heat generated by the megafires can loft fire emissions high in the troposphere, a process which is not captured by the climatological profiles. For low fire emissions, the climatological method shows high biases in injected emission fluxes above the PBL due to inaccurate climatological plume profiles in north and central of Australia.

Figure 2c compares the plume injection fractions above the PBL forecast by the random forest model against those derived from the MISR plume profiles and daily mean PBL height. These samples are from the test dataset, which is independent from the data used for random forest training. Our random forest model generally captures the plume injection fractions compared to the MISR observations, with a normalized mean bias (NMB) of 1.3% and a RMSE of 0.22. The model explains 53% of the variance in the injection fractions derived from MISR, with overestimates at the low end and underestimates at the high end of the distribution, which can be partly attributed to systematic biases associated with ensemble-tree machine learning regressions (Zhang and Lu, 2011; Belitz and Stackelberg, 2021). In addition, we include only 191 records of plume height retrievals in November 2019, most of which are associated with large injection fractions. This relatively limited plume record may not have been adequate to predict the plume behavior of intense fires with confidence. We also compare the observed the model fire emission fluxes injected above the PBL (Figure 2d). Here our model successfully captures 88% of the variance in the

observed fluxes in the test dataset, with NMB of 10%. The high model bias for small injection fractions leads to only a slight overestimate of smoke fluxes above the boundary layer, as such small fractions are generally associated with low mass fluxes.

**3.3 Predictor importance for predicting plume injection fractions by random forest**

Figure 3 shows the importance of each predictor from the random forest model, which is calculated as described in Section 2.2.3. Larger values indicate greater importance. We find that the important variables include daily mean PBL height (PBLH), air temperature at 2 m (T2), meridional wind speed at 10 m (V10), and the corresponding fire emissions (EMIS). The first three variables are highly related to ambient atmospheric stability (Mohan and Siddiqui, 1998) and fire behavior (Schroeder and Buck, 1970). Wildfire smoke disperses more under higher PBL heights and unstable atmospheric conditions, which in turn may be affected by the movement of warmer air into the area near the surface or colder air into the area aloft (Schroeder and Buck, 1970). Thermal advection can be highly related to the meridional wind speed. Fire emissions implicitly reflect both the fire intensity and fuel load. The combined effects of these factors thus influence the degree to which the smoke plume is injected above the boundary layer. The maximum FRP within the plume is relatively less important predicting injection fractions above the PBL, consistent with previous studies which documented the weak correlation between FRP and injection height (Kahn et al., 2007; Val Martin et al., 2012). This weak correlation can be traced in part to clouds or smoke obscuring fires from satellite detection or to incomplete knowledge of the local temperature profile. Previous studies have attempted to directly correlate plume injection heights with FRP observations. However, the relationship between observed FRP and the convective heat flux driving the plume rise depends in large part on the local temperature profile which may not be well known (Kahn et al., 2007). In addition, the satellite pixels may be only partly filled by fire, leading to an underestimate of the heat flux driving plume rise.

**3.4 Comparison of plume injection fractions calculated by random forest and climatological injection profiles**

Figure 4 illustrates the spatial distributions of annual mean fractions of total fire emissions injected above the PBL in each grid cell, calculated by the two methods during April to January of the next year, averaged over 2009 to 2020. (This timeframe includes the fire seasons of both northern and southeastern Australia.) The injection fractions derived from the climatological injection profiles range from 10 to 50% across much of northern Australia. In contrast, the random forest method predicts strong lofting of smoke in more limited regions in FNQ and in the eastern area of the Top End, where about 30% of total fire emissions are injected

into the free troposphere. Elsewhere in northern Australia, the random forest method yields injection
fractions above the PBL of only 10% to 20% of total fire emissions. In the coastal areas of southeastern
Australia, the climatological method estimates that less than 40% of fire emissions are lifted above the
boundary layer, while the random forest method predicts that the injection fractions account for 40-60%.
Put another way, the climatological method predicts that about ~18% less OC emissions remain within the
PBL on average over northern Australia, compared to the random forest method (Figure 4c). Over
southeastern Australia, the situation is reversed, with INJ-CLIM predicting ~14% more emissions within
the PBL on average than INJ-RF (Figure 4c). In southeastern Australia, we find that the spatial distribution
of large plume injection fractions predicted by random forest is highly correlated with that of high OC fire
emissions in coastal areas (Figure 1). Given the good match of these injection fractions with MISR
observations, we conclude that our random forest model better captures extreme wildfire events compared
to the climatological method due to inclusion of daily fire emissions and FRP as predictors.

Figure 5 compares the estimated monthly mean OC fire emissions within the PBL using the two methods
in northern Australia and southeastern Australia during their respective fire seasons from 2009 to 2020. In
northern Australia, the climatological method predicts an average 17.6 Gg month$^{-1}$ of fire-emitted OC lifted
above the PBL, or 45% of the total OC fire emissions (39 Gg month$^{-1}$) during the fire season (April to
December). In contrast, the random forest method predicts just 10.6 Gg month$^{-1}$ of fire-emitted OC lifted
above the PBL, or just 27% of total OC fire emissions on average (Figure 5c). Although there is large
interannual variation of monthly mean total OC fire emissions, ranging from 18.6 Gg month$^{-1}$ to 62.9 Gg
month$^{-1}$, neither method shows a long-term trend of plume injection fractions in northern Australia over the
last decade. In southeastern Australia, the interannual changes in both fire emissions and plume injection
fractions estimated by INJ-CLIM and INJ-RF methods are more pronounced from 2009 to 2020 compared
to those in northern Australia. This is due to the dramatic changes in total amounts of wildfires and fire
intensity in this region. In 2019, monthly mean total OC fire emission during the extreme fire season is 481
Gg month$^{-1}$, significantly higher than in other years, in which total OC fire emissions average just 13.7 Gg
month$^{-1}$ (Figure 5b). In addition, Figure 5d shows that 48% of total OC fire emissions are released above
the PBL in 2019 forecast by the random forest model, much larger than the injection fraction (30%)
estimated by climatological method. During other years, the injection fractions estimated by the two
methods are similar, with mean values of 33.5% (climatological injection profiles) and 37.9% (random
forest model). On average across southeastern Australia, the climatological method and random forest
method yield 31% and 45%, respectively, of total fire emissions rising above the PBL from 2009 to 2020.

**4 Application to smoke PM$_{2.5}$ and their contributions to air quality across Australia during fire seasons**

**4.1 Validation of total PM$_{2.5}$ simulated by sensitivity experiments**

We apply the resulting plume injection fractions quantified by the two methods to smoke PM$_{2.5}$ simulations using the STILT model at 12 receptors in nine key cities with large populations during the fire seasons from 2009 to 2020. Figure 6 shows the receptor locations, which are located in the northern and southeastern Australia. The three sensitivity experiments (CTL, INJ-CLIM, and INJ-RF) are driven by fire emissions with different injection scenarios, as described in Section 2.3.2 and Table 2. We rely on the ground-based measurements of total PM$_{2.5}$ concentrations and the estimated non-fire PM$_{2.5}$ concentrations (described in Section 2.4) to test the accuracy of our two approaches for quantifying the plume injection fractions and their impacts on long-term smoke exposure. Total modeled PM$_{2.5}$ is assumed to consist of smoke PM$_{2.5}$ and non-fire PM$_{2.5}$. Table S1 shows the statistics of annual mean surface total PM$_{2.5}$ simulated by the three sensitivity experiments, compared to total PM$_{2.5}$ observations at 12 receptors during the fire seasons over the last decade. The three experiments reproduce the interannual variability of PM$_{2.5}$ concentrations with temporal correlation coefficients ranging from 0.54 to 0.99, except for the receptor Footscray in Melbourne. The NMBs and RMSEs between the simulations and observations vary depending on the injection scenario, ranging from -32.2% to 19% for NMBs and 0.69 μg m$^{-3}$ to 7.0 μg m$^{-3}$ for RMSEs. At most sites, the results from the INJ-RF and INJ-CLIM experiments yield relatively lower RMSEs and NMBs against observations compared to the CTL experiment, indicating the importance of considering plume injection heights on modeling smoke concentration for exposure estimates in Australia. However, there are large biases in simulated total PM$_{2.5}$ concentrations from the INJ-RF experiment compared to the observations in Gladstone, Brisbane, Wollongong, Canberra, Newcastle (Wallsend), and Albury. In Gladstone, Wollongong, and Albury, we also find the low biases in simulated total PM$_{2.5}$ concentrations from the CTL experiment, indicating that the total fire emissions from the original GFED 4.1s or the estimated non-fire PM$_{2.5}$ concentrations may be underestimated. The inclusion of plume injection in the INJ-RF and INJ-CLIM experiments thus aggravate low biases in simulated smoke PM$_{2.5}$ concentrations over the three cities. In Brisbane, Canberra, and Newcastle (Wallsend), the low biases are relatively significant during the high-fire years of 2009 and 2019. We speculate that these biases arise from neglect in our model setup from downward mixing of smoke plumes in remote regions (Section 2.3.2).

Figure 7 compares the time series of total PM$_{2.5}$ concentrations simulated by the three experiments against the surface measurements at six representative sites in northern and southeastern Australia during the fire season in 2019-2020. We use the 10-day averages of simulated total PM$_{2.5}$ concentrations to reduce the

impacts of weather conditions on day-to-day variability of non-fire $PM_{2.5}$, which is set to a constant value for each year at each receptor in our study, and to smooth out the response of smoke $PM_{2.5}$ to modeled fluctuations in fire activity. These fluctuations depend on the daily scale factors provided by GFED 4.1s

and are somewhat uncertain. The three experiments successfully capture the remaining variability of $PM_{2.5}$ with temporal correlation coefficients ranging from 0.59 to 0.93, indicating that smoke $PM_{2.5}$ contributes much of the synoptic-scale variation of total $PM_{2.5}$ in these cities during the fire season. Compared to the CTL experiment, the INJ-RF experiment significantly reduces the overestimate of total $PM_{2.5}$ concentration in Newcastle (77.5% to 9.2%), Sydney (27.9% to -6.3%), and Canberra (47% to -8.2%), three cities which

are close to the most extreme fire events of 2019-2020. In particular, compared to results from the INJ-CLIM experiment, the peak values of total $PM_{2.5}$ simulated by INJ-RF experiment agree best with observations in Newcastle and Sydney during the megafires of November to Mid-December, with the lowest NMBs of 31% and -5.0%. In Melbourne, three experiments capture fire events from December to January with temporal correlation coefficients over 0.92. However, the simulated total $PM_{2.5}$ concentrations

are underestimated with NMBs ranging from -28.2% to -20.9% in all three experiments. Again, the peak values of smoke $PM_{2.5}$ concentrations are also unable to be captured by CTL experiment, which can be traced to the low biases from the fire emission inventory.

We further validate the time series of simulated and observed total $PM_{2.5}$ concentrations at all receptors, averaged over their respective observation periods (Figure S2). Table 3 shows the statistics for daily mean

$PM_{2.5}$ concentrations simulated by CTL, INJ-CLIM, and INJ-RF experiments, compared to the ground-based observations at 12 receptors. These average concentrations reveal the long-term smoke exposure at these receptors. The three model experiments successfully reproduce the time series of daily $PM_{2.5}$ at most receptor cities except for Wollongong and Melbourne, with temporal correlation coefficients ranging from 0.4 to 0.93. In Wollongong and Melbourne (Footscray), where $R$=0.27 and 0.25, smoke $PM_{2.5}$ contributes

only 10% and 5% of total $PM_{2.5}$ from 2009 to 2020 (Figure 6). The daily variations of $PM_{2.5}$ in the two cities are thus mainly affected by weather conditions and anthropogenic emissions in some low-fire years, and our model is unable to capture this. Compared to INJ-CLIM, INJ-RF yields higher correlation coefficients and smaller RMSEs at most receptors, indicating that INJ-RF better captures the daily variability and peak values of total $PM_{2.5}$ concentrations during the fire seasons. However, INJ-RF improves

the NMBs only in Darwin, Sydney (Liverpool), Melbourne (Alphington), and Newcastle. In other receptors, the total $PM_{2.5}$ concentrations are more underestimated in the INJ-RF experiment than in INJ-CLM, possibly due in part to the neglect in our model setup of downward mixing of smoke in remote regions.

**4.2 Impacts of plume injection heights on annual mean smoke exposure in northern and southeastern Australia**

Figure 8 compares the annual mean smoke $PM_{2.5}$ simulated by STILT and background $PM_{2.5}$ against ground-based observations of total $PM_{2.5}$ at six representative sites in Australia over the last decade. Figure S3 shows the results in other six sites. The differences in simulated total $PM_{2.5}$ are driven by different plume injection scenarios and derived smoke $PM_{2.5}$ concentrations. Figure 9 shows the mean sensitivity footprints at six representative cities during the fire seasons from 2009 to 2020. The panels indicate the time-average

source regions of the air masses reaching these receptors within 120 hours. When these air masses originate from burning regions, the surface $PM_{2.5}$ concentrations at the receptors show enhancements of smoke $PM_{2.5}$. In contrast, the impacts of wildfire smoke are quite small when the upwind source regions are over the ocean or non-burning areas.

     Darwin is the capital city of Northern Territory located in the Top End, with long fire seasons from April

to December.. We find that this city is significantly affected by biomass burning in the Top End, where the mean sensitivity footprints are higher than $1 \times 10^{-3}$ ppm $\mu mol^{-1}$ $m^2$ s (Figure 9a). In the CTL experiment, simulated total $PM_{2.5}$ is 16.7% higher than the observations on average, with overestimates increasing to 31%-47% during the years with stronger fire emissions (e.g., 2011, 2012, and 2015). However, the INJ-CLIM experiment underestimates the simulated total $PM_{2.5}$ by 18.0%, indicating a likely overestimate of

fire emissions injected above the PBL. One possible reason for this overestimate can be traced to the inaccuracies in the input data and the semi-empirical parameterization (Rémy et al., 2017). Based on Sofiev et al. (2013), plume injection height is proportional to the PBL height, which is usually large in northern Australia compared to other regions, leading to a higher injection fraction of fire emissions above the PBL. In the INJ-RF experiment, the mean simulated total $PM_{2.5}$ concentrations are in best agreement with the

surface measurements with a NMB of -2.5% averaged from 2011 to 2020. This finding demonstrates the importance of considering the plume injection heights of smoke $PM_{2.5}$ during the severe fire seasons, as well as the regional differences in fire dynamics.

     Gladstone is located on the east coast of Queensland and is influenced by burning in eastern Australia (Figure 9b). We find that annual mean wildfire contributions to total $PM_{2.5}$ varies greatly at this site, from

2% to 36% over the last decade based on the results of INJ-RF experiment. Smoke $PM_{2.5}$ concentrations account for less than 10% of the total $PM_{2.5}$ in Gladstone during 2009 to 2010, 2012, and from 2014 to 2017. During low-fire years, the low biases in simulated total $PM_{2.5}$ are likely caused by an underestimate of background $PM_{2.5}$ concentrations from anthropogenic emissions. During the high-fire years of 2013 and 2018, the INJ-RF experiment performs better than the CTL experiment, with negligible NMBs of 0.8% and

6.3%. In 2011 and 2019, however, INJ-RF underestimates total $PM_{2.5}$ by 22% and 29.5%. The significant

underestimates of total $PM_{2.5}$ can be partially attributed to the low biases in the fire emission inventory, which also leads to 15% and 18% underestimates of total $PM_{2.5}$ from the CTL experiment. Another reason may be neglect in our model setup of downward mixing of smoke far from the source regions. During the fire seasons in 2011 and 2019, Gladstone experiences the impacts of smoke from both local and remote burning regions in eastern coastal area.

In southeastern Australia, we find similar trends in annual mean smoke $PM_{2.5}$ concentrations and their contributions to total $PM_{2.5}$ in Brisbane, Newcastle, and Sydney (Figure 8c, 8d, and 8e). These sites are sensitive to the fire emissions in eastern coastal areas. Figure 9c, 9d, and 9e show that general upwind regions to the three cities are over both land and ocean from 2009 to 2020. During the 2019 high-fire year, the CTL experiment greatly overestimates total $PM_{2.5}$ concentrations by 73% and 30% in Newcastle and Sydney, respectively. Annual mean smoke $PM_{2.5}$ in the CTL simulation is even larger than observed total $PM_{2.5}$ in Newcastle, which suggests that a considerable fraction of fire emissions is released above the PBL in the source regions upwind of this city. The CTL experiment also overestimates total $PM_{2.5}$ concentrations by 30% to 54% in Brisbane during 2010, 2012 to 2013, and 2018, and by 15% to 29% in Sydney from 2012 to 2013. The contributions of smoke $PM_{2.5}$ to total $PM_{2.5}$ ranges from 20% to 45% during these years. The INJ-CLIM experiment partially improves the modeled smoke $PM_{2.5}$ compared to the CTL experiment by introducing the climatological plume injection of fire emissions, but the climatological injection profiles are unable to accurately reflect the fire emissions injections during severe fire seasons. In contrast, the INJ-RF experiment best matches the smoke $PM_{2.5}$ simulations in the cities near the burning regions during these high-fire years. For example, INJ-RF and INJ-CLIM reduce the large CTL overestimate of total $PM_{2.5}$ concentrations in Newcastle from 73% to 6.6% (INJ-RF) and 25.5% (INJ-CLIM) during 2019. But in remote downwind regions, both INJ-RF and INJ-CLIM experiments underestimate the smoke $PM_{2.5}$ concentrations in 2019, probably due to neglect in our model of downward mixing of fire plumes from high altitudes. The INJ-CLIM experiment estimates more fire emissions remaining within the PBL, which yields a smaller low bias in Brisbane. INJ-RF yields NMBs of total $PM_{2.5}$ ranging from 1.5% to 24.3% compared to observations in Sydney and Brisbane during 2010, 2012, and 2013, smaller than the NMBs (6.3% to 54%) in the CTL experiments. During other low-fire years when smoke $PM_{2.5}$ contributes less than 10% of total $PM_{2.5}$, the simulated smoke $PM_{2.5}$ concentrations from INJ-CLIM and INJ-RF experiments are similar.

Figure 8f shows the results of three simulations in Melbourne, where the fire seasons have significantly varied during the austral summer (December to the following January) over the last decade. The fire season in Melbourne is shifted later in the year compared to New South Wales and Queensland. The sensitivity footprint of Figure 9f illustrates that Melbourne is mainly affected by southwesterly winds, which may bring marine air onshore. Thus, fire emissions from southeastern Australia contribute just 1% to 8% of total $PM_{2.5}$ concentrations at this site, except for the high-fire years 2009, 2011, and 2018-2019, when these

contributions range from 15% to 22%. In the high-fire years, we also find a modest improvement in simulated total $PM_{2.5}$ from the INJ-RF experiment (2009: NMB = 4.4%; 2018: NMB = 11.6%), compared to the results from the INJ-CLIM experiment (2009: NMB = 13.5%; 2018: NMB = 34.5%).

**4.3 Contributions of long-term smoke $PM_{2.5}$ to regional air quality**

    We next calculate the ratios of simulated smoke $PM_{2.5}$ concentrations from the INJ-RF experiment to

observed total $PM_{2.5}$ concentrations averaged in respective observation periods at 12 receptors to quantify the long-term contributions of wildfires in key Australian cities (Figure 6). Here we use observations for total $PM_{2.5}$ concentrations in these ratios, rather than the sum of modeled smoke and non-smoke $PM_{2.5}$, as the observations provide greater certainty. Figure 10 shows the annual mean contributions of smoke $PM_{2.5}$ at all receptors during the last decade. On average, the long-term smoke $PM_{2.5}$ accounts for 5% to 52% of

total $PM_{2.5}$ across all receptors during the fire seasons. Smoke $PM_{2.5}$ contributes the most in Darwin, accounting for 35% to 74% from 2011 to 2020. In the seven receptors located in the eastern coastal area, mean smoke $PM_{2.5}$ contributions range from 9% to 23% with large interannual variations. For example, at the Liverpool site in Sydney, smoke $PM_{2.5}$ accounts for 4% to 38% of total $PM_{2.5}$, and as much as 33% to 38% during the intense 2013 and 2019 fire seasons. In other inland receptors and Melbourne, the annual

smoke $PM_{2.5}$ contributions are usually less than 10%, but the contributions rise as high as 20% during high-fire years of 2009, 2011, and 2019 in southeastern Australia. The large mean smoke contribution (73%) in Florey, a suburb of Canberra, is caused by the extreme fire events in 2019. The smoke contributions are less than 5% in other years from 2014 to 2020.

    Figure S4 shows the contributions of wildfires to total $PM_{2.5}$ during the 2019-2020 fire season, when

extreme fire events occurred in southeastern Australia. We find that in northern cities, the smoke $PM_{2.5}$ contributions are consistent with those in the long-term averages (Figure 6). But in some densely populated cities in southeastern Australia, the contributions of smoke $PM_{2.5}$ significantly increase during this time frame, from 17% to 38% in Sydney, 17% to 54% in Newcastle, 40% to 73% in Canberra, and 9% to 15% in Melbourne. Our results highlight the short-term impacts that this severe wildfire season had on regional

air quality.

    At most sites examined in Australia, smoke $PM_{2.5}$ drives the seasonal variations of total $PM_{2.5}$. Figure S5 shows the monthly mean contributions of smoke $PM_{2.5}$ at six representative sites over the last decade. In Darwin, mean smoke $PM_{2.5}$ contributions rise to over 50% from May to August, but fall to less than 20% from November to December. This seasonality is consistent over the last decade and can be traced to the

influence of the Australian monsoon, as described in the Introduction. The wildfires in the Top End and FNQ usually last from April to December. From April to August, a high-pressure system is typically located in southern Australia. Southeasterly winds from this area are warm and dry, bringing smoke from burning

regions in the Top End to Darwin. After September, the monsoon carries warm and moist oceanic air into Darwin from the northwest, limiting the impact of wildfire smoke emitted over the Top End and FNQ on

air quality into the city. The STILT model usually yields a better performance capturing the patterns of sensitivity footprints due to the reliable meteorological variables provided by GDAS and GFS. In southeastern Australia, the peak time of smoke $PM_{2.5}$ contributions to total $PM_{2.5}$ are from August to the following January, lagging that in northern Australia. In Gladstone, smoke $PM_{2.5}$ accounts for less than 5% during April to July as a result of low fire emissions in the upwind eastern coastal area. During August to

December, mean smoke $PM_{2.5}$ contributions in this city increase from 8% to 16% due to more frequent fire activity in the region. In Brisbane, Newcastle, Sydney, and Melbourne, the peak fire periods occur during October to January, when summer heat dries out the forest and grasses that fuel the fires. These four cities then become vulnerable to the threat of wildfires smoke, with mean contributions to total $PM_{2.5}$ ranging from 13% to 25%. However, the wildfire events in southeastern Australia experience large interannual

variability, resulting in variable spatiotemporal distributions of fire emissions during fire seasons over the last decade. Air quality in the other five cities of southeastern Australia that we examine are affected by surface air fluxes from both land and ocean. The day-to-day variability of sensitivity footprints in these receptors are pronounced, which may be challenging for the STILT model to accurately reproduce.

**5 Discussion and conclusion**

We have developed two approaches to quantify the plume injection fractions above the PBL over Australia during the fire seasons from 2009 to 2020, with the goal of improving estimates of smoke $PM_{2.5}$ exposure in cities downwind of fires. Both methods estimate the daily fraction of smoke plumes injected above the PBL. The climatological approach is based mainly on the climatological monthly mean injection profiles from IS4FIRES and daily injection heights compiled by the GFAS emission inventory. For the

second approach, we train a random forest model to predict the daily plume injection fractions, using plume heights derived from MISR observations, assimilated meteorology, and fire information from MODIS and GFED 4.1s. The climatological method can explain 76% of variances in daily mass flux of fire emissions injected above the PBL derived from MISR, but it underestimates the mean value of this flux by 63% in the test dataset. A likely reason for this weakness is that the climatological injection profiles cannot capture

the spatiotemporal variability of plume injection fractions. The resulting random forest model, in contrast, more successfully reproduces the mass flux of fire emissions injected above the PBL, with an $R^2$ of 0.88 and NMB of 10%, compared to MISR observations. To quantify the impact of plume injection fractions on smoke air quality, we then apply total fire emissions to STILT together with the plume injection fractions that remain within the PBL.

We find that characterization of the plume injection fractions greatly affects estimates of the surface daily smoke $PM_{2.5}$ in northern and southeastern Australia, especially during severe fire seasons when intense heat from fires can loft smoke high in the troposphere or even to the stratosphere. The random forest model predicts plume behavior that best agrees with observed surface $PM_{2.5}$, especially over the receptors near the burning regions during most high-fire years. For example, in northern Australia, when assuming that all

fire emissions are released within the PBL, STILT generates total $PM_{2.5}$ concentrations ~16% higher than surface observations on average in Darwin during the last decade. Using the climatological method, however, we estimate that ~45% of smoke emissions rises above the PBL at Darwin, while the random forest method estimates just 27%. Applying these plume injection fractions to STILT reduces the NMBs between simulated and observed total $PM_{2.5}$ concentrations to -18% for the climatological method and -2.5%

for the random forest method.

In southeastern Australia, we find that both fire frequency and injection fractions significantly vary over the last decade. During the severe fire season of 2019, the random forest method predicts that 48% of smoke plume mass rises above the PBL, much higher than the 30% estimated by climatological method. In Sydney and Newcastle, these two methods generate surface concentrations in better agreement with observations than the control simulation, with NMBs of -4.5% (INJ-RF) to -7.0% (INJ-CLIM) in Sydney and 6.6% (INJ-

RF) to 25.5% (INJ-CLIM) in Newcastle. However, neither method can quantify the possible downward mixing of fire smoke plumes in downwind regions and the subsequent impact on surface air, a process which may be especially important during more severe fire seasons when intense heat lofts greater quantities of smoke above the PBL in source regions. In Brisbane, Gladstone, and Melbourne, the INJ-RF

method leads to more pronounced underestimates of surface $PM_{2.5}$ concentrations compared to INJ-CLIM, perhaps because of this shortcoming.

We further quantify the long-term contributions of smoke $PM_{2.5}$ in key Australian cities based on the simulations with the INJ-RF plume injection scenario. Results show that smoke $PM_{2.5}$ accounts for 5% to 52% of the total $PM_{2.5}$ during the fire seasons from 2009 to 2020. In most cities in southeastern Australia,

we find large interannual variations of smoke $PM_{2.5}$ contribution to total $PM_{2.5}$, ranging from 1% to 73%, suggesting the vulnerability of this region to infrequent but extreme smoke events. For example, during the 2019-2020 "Black Summer," smoke accounts for 38% of total $PM_{2.5}$ in Sydney, 54% in Newcastle, and 73% in the Canberra, indicating the vulnerability of populations living close to the intense wildfires.

The machine learning approach (INJ-RF) has two main limitations. First, the relatively short fire season

and interannual variability of fire activity in southeastern Australia means that fewer MISR records are currently available to train and test the INJ-RF model. Digitizing more smoke plume records from MISR, a laborious process, could enhance the training and testing of the INJ-RF model. Future studies could then train the random forest models separately in the two regions – northern and southeastern Australia – and

identify the drivers for each region. Second, as noted above, the STILT model cannot capture downward mixing of smoke away from source regions. Future studies could explore the impacts of long-range transport and downward mixing of fire emissions on surface smoke concentrations by applying the estimated injection fractions to 3-D chemical transport models.

Climate change is projected to increase fire frequency in many regions worldwide (Abatzoglou and Williams, 2016; Di Virgilio et al., 2019; Canadell et al., 2021), and knowledge of plume behavior is essential to accurately quantify the resulting smoke exposure and health impacts. Our random forest model for calculating plume injection fractions promises to improve assessment of surface smoke concentrations downwind of fires. The model can predict the daily plume injection fractions above the PBL at $0.25° \times 0.25°$ horizontal resolution or higher, depending on the spatial resolution of the fire emission inventory. Thus, this approach predicts plume behavior at a higher spatiotemporal resolution than the climatological approach used here. Our method can be easily applied to other regions and implemented in 3-D chemical transport models, which can better represent the long-term transport of smoke in vertical layers than can Lagrangian plume models like STILT. The accuracy of the random forest predictions may be further improved once more satellite retrievals of fire plume heights become available for model training. The utility of the machine learning approach can also be explored in regions where satellite observations of plume heights are missed due to cloud obscuration or inadequate overpass frequency.

**Code availability.** The STILT model is open-source and is available from https://uataq.github.io/stilt/#/ (last access: 17 June 2023). Other source code is available from

https://github.com/fengx7/HEI_Australian_wildfires/tree/main.

**Data availability.** The data is available from the authors upon request.

**Author contributions.** LJM and MLB designed and oversaw the project. XF developed the methods for

quantifying the plume injection fractions, performed the simulations and analysis, and wrote the manuscript. TL assisted in performing the STILT simulations and calculating background $PM_{2.5}$. JAF assisted in designing the sensitivity experiments and in processing the $PM_{2.5}$ observation data. MVM assisted in processing MISR observation data and using the MINX software. All authors contributed to the manuscript.


**Competing interests.** The authors declare no competing interests.

**Acknowledgements.** This work is supported by the Health Effects Institute (HEI), an organization jointly funded by the United States Environmental Protection Agency (EPA) (Assistance Award No. CR-

83998101) and certain motor vehicle and engine manufacturers. Computational resources are provided by the Faculty of Arts & Sciences Research Computing (FASRC) Cannon compute cluster at Harvard University.

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

Table 1. Predictors and target variable for the random forest model in this study

| **Target variable** | | |
| --- | --- | --- |
| **Short name** | **Description (unit)** | **Data source** |
| Plume injection fractions | Daily plume injection fractions above the PBL (%) [a] | MISR Plume Height Project 2 (1.1 km for blue band, 275 m for red band; daily); MERRA-2 (0.5° latitude × 0.625° longitude, 1-hour) |
| **Predictors** | | |
| **Short name** | **Description (unit)** | **Data source (spatial & temporal resolution)** |
| LANDUSE | Land use classification (unitless) | MODIS Land Cover Climate Modeling Grid Version 6 (MCD12C1) (0.05° latitude × 0.05° longitude, yearly) |
| PBLH | Daily mean PBL height (m) | MERRA-2 (0.5° latitude × 0.625° longitude, 1-hour) |
| T2 | Daily mean air temperature at 2 m (K) | MERRA-2 (0.5° latitude × 0.625° longitude, 1-hour) |
| RH | Daily mean surface relative humidity (%) | MERRA-2 (0.5° latitude × 0.625° longitude, 3-hour) |
| U10 | Daily mean U-wind at 10 m (m s$^{-1}$) | MERRA-2 (0.5° latitude × 0.625° longitude, 1-hour) |
| V10 | Daily mean V-wind at 10 m (m s$^{-1}$) | MERRA-2 (0.5° latitude × 0.625° longitude, 1-hour) |
| PRECIP | Daily total precipitation (kg m$^{-2}$ s$^{-1}$) | MERRA-2 (0.5° latitude × 0.625° longitude, 1-hour) |
| EMIS | Daily mean OC biomass burning emissions (kg m$^{-2}$ s$^{-1}$) | GFED 4.1s (0.25° latitude × 0.25° longitude, daily) |
| LON | Longitude of the biomass burning emission grid cell (degree) | GFED 4.1s (0.25° latitude × 0.25° longitude, daily) |
| LAT | Latitude of the biomass burning emission grid cell (degree) | GFED 4.1s (0.25° latitude × 0.25° longitude, daily) |
| FRP | Maximum fire radiative power within the plume (MW) | MODIS/Terra Thermal Anomalies/Fire 5-Min L2 Swath 1km V061 (MOD14) (2030 km along swath × 2300 km across swath, 5-minute) |

[a] Fraction of plume pixels injected above the PBL within plume perimeter. Detailed calculation is described in Eq. (1) Section 2.2.1.


Table 2. Configurations of STILT experiments in this study.

| Experiments | Case CTL[a] | Case INJ-CLIM[b] | Case INJ-RF[c] |
|---|---|---|---|
| Domain (spatial resolution) | 112.0° E to 165.5° E, 45.5° S to 9.5° S (0.25° × 0.25°) | | |
| Simulation period | April 1st to December 31st (for 2 receptors in northern Australia) August 1st to December 31st (for 10 receptors in southeastern Australia) | | |
| Backward simulation time (start time) | 120-hr (00:00:00 AEST, UTC+10) | | |
| Air parcel number | 1000 | | |
| Meteorological data, spatial resolution (years) | GDAS, 0.5° × 0.5° (2009 to 2018) GFS, 0.25° × 0.25° (2019 to 2020) | | |
| STILT algorithm | Gaussian kernel density estimation w/o the hyper-near field vertical mixing depth correction | | |
| Fire emissions | No scaling | Scaling with injection fraction from the climatological method | Scaling with injection fraction from the random forest method |

[a] Case CTL represents the control experiment that assumes total fire emissions are released within the PBL.
[b] Case INJ-CLIM represents the first sensitivity experiment that assumes some fire emissions are injected above the PBL based on injection fractions from the climatological method.
[c] Case INJ-RF represents the second sensitivity experiment that assumes some fire emissions are injected above the PBL based on injection fractions from the random forest method.

Table 3. Statistics for daily mean PM$_{2.5}$ concentrations simulated by CTL, INJ-CLIM, and INJ-RF experiments, compared to the ground-based observations at 12 receptors. The daily mean concentrations are calculated over each receptor's observing period. Shown are the temporal correlation coefficients $R$, NMBs, and RMSEs of daily total PM$_{2.5}$ concentrations compared to the surface measurements.

| Cities (site) | Observation periods (Locations) | $R$ [a] | | | NMB | | | RMSE (µg m$^{-3}$) | | |
|---|---|---|---|---|---|---|---|---|---|---|
| | | CTL | INJ-CLIM | INJ-RF | CTL | INJ-CLIM | INJ-RF | CTL | INJ-CLIM | INJ-RF |
| Darwin [b] (Palmerston) | 2011-2020 (130.94°E, 12.50°S) | 0.76 | 0.76 | 0.76 | 17.1% | -17.8% | -2.1% | 5.9 | 3.9 | 4.0 |
| Gladstone [c] (South Gladstone) | 2009-2020 (151.27°E, 23.86°S) | 0.57 | 0.55 | 0.55 | -5.4% | -11.0% | -11.2% | 1.9 | 1.7 | 1.7 |
| Brisbane [c] (Springwood) | 2009-2020 (153.13°E, 27.61°S) | 0.24 | 0.17 | 0.40 | 13.2% | 2.2% | -4.8% | 2.2 | 1.9 | 1.2 |
| Newcastle [d] (Wallsend) | 2009-2020 (151.66°E, 32.89°S) | 0.53 | 0.48 | 0.52 | 16.1% | 2.3% | -6.0% | 6.6 | 3.9 | 2.4 |
| Sydney [d] (Liverpool) | 2009-2020 (150.90°E, 33.93°S) | 0.40 | 0.38 | 0.37 | 14.8% | 6.8% | 0.4% | 4.8 | 3.4 | 2.8 |
| Wollongong [d] (Wollongong) | 2009-2020 (150.88°E, 34.41°S) | 0.27 | 0.28 | 0.27 | -6.3% | -11.6% | -14.7% | 1.5 | 1.4 | 1.6 |
| Melbourne [e] (Footscray) | 2009-2020 (144.87°E, 37.80°S) | 0.25 | 0.25 | 0.25 | -9.5% | -10.1% | -14.6% | 4.6 | 4.6 | 3.6 |
| Melbourne [e] (Alphington) | 2009-2020 (145.03°E, 37.77°S) | 0.41 | 0.39 | 0.40 | 23.7% | 22.9% | 14.4% | 2.0 | 2.0 | 1.2 |
| Albury [d] (Albury) | 2017-2020 (146.93°E, 36.05°S) | 0.93 | 0.93 | 0.93 | -22.2% | -23.7% | -31.7% | 5.6 | 5.8 | 7.3 |
| Canberra [f] (Florey) | 2014-2020 (149.04°E, 35.22°S) | 0.67 | 0.63 | 0.68 | 19.3% | -8.6% | -16.0% | 22.5 | 17.7 | 15.3 |
| Sydney [d] (Prospect) | 2014-2020 (150.91°E, 33.79°S) | 0.72 | 0.69 | 0.71 | 7.5% | -1.2% | -7.2% | 2.2 | 1.5 | 1.4 |
| Newcastle [d] (Newcastle) | 2014-2020 (151.75°E, 32.93°S) | 0.59 | 0.50 | 0.55 | 28.9% | 9.2% | -1.6% | 6.9 | 4.0 | 2.7 |

[a] Temporal correlation coefficient between the observed and simulated annual mean total PM$_{2.5}$ concentrations during the fire

seasons (April to December for Darwin and Gladstone; August to December for other cities).

[b] Observation data source: Northern Territory Environment Protection Authority (http://ntepa.webhop.net/NTEPA/Default.ltr.aspx)

[c] Queensland Government Open Data Portal (https://apps.des.qld.gov.au/air-quality/download/)

[d] New South Wales Department of Planning and Environment (https://www.dpie.nsw.gov.au/air-quality/air-quality-data-services/data-download-facility)

[e] Victoria Environment Protection Authority (https://www.epa.vic.gov.au/for-community/airwatch)

[f] Australian Capital Territory Government Open Data Portal (https://www.data.act.gov.au/Environment/Air-Quality-Monitoring-Data/94a5-zqnn)

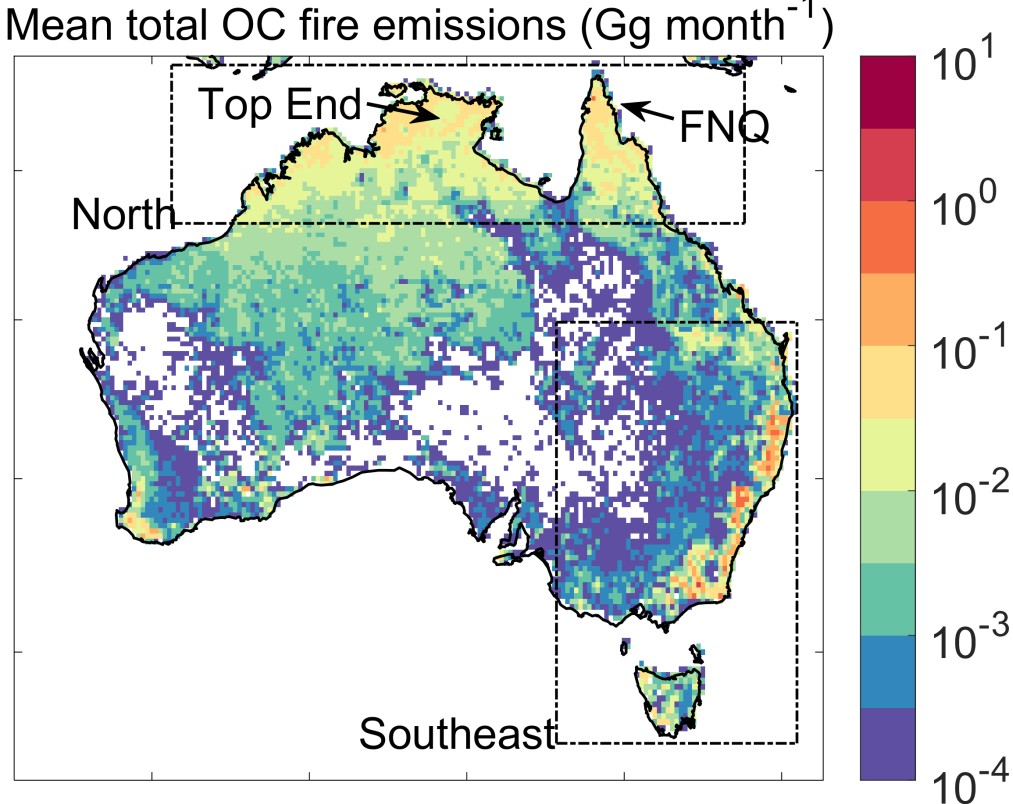

Figure 1. Spatial distributions of annual mean total OC fire emissions during April to the following January from 2009 to 2020, in units of Gg month⁻¹. The dashed boxes represent the northern and southeastern Australia in this study. Also shown are the locations of the Top End and Far North Queensland (FNQ).

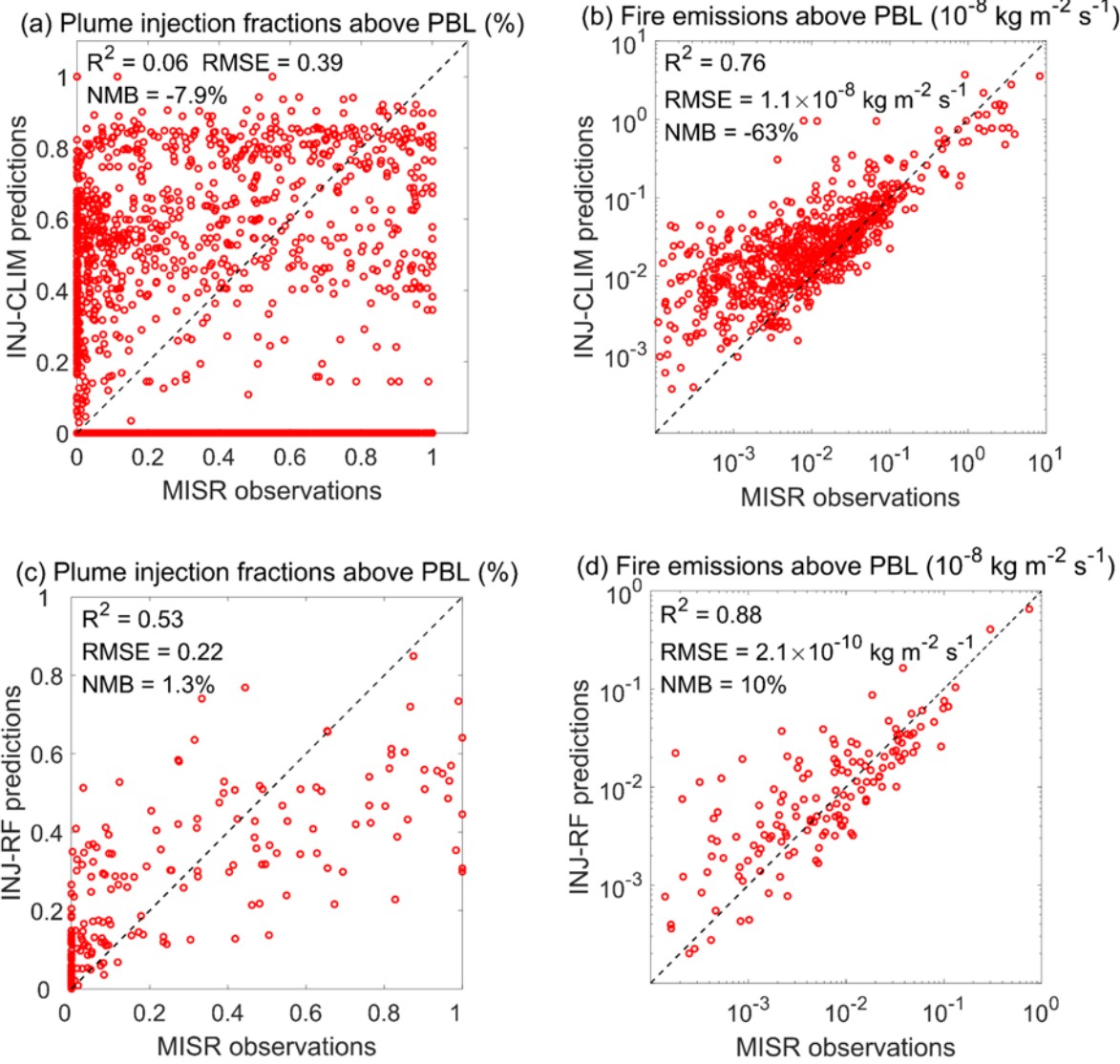

Figure 2. Scatter plots of plume injection fractions above the PBL calculated by (a) climatological method and by (c) random forest, compared to the MISR derived plume injection fractions. Scatter plots of mass fluxes of fire emissions injected above the PBL estimated by injection fractions from (b) climatological method and (d) random forest, compared to MISR observations, in units of kg m$^{-2}$ s$^{-1}$. Also shown are the $R^2$ values and the RMSEs and NMBs between the predictions and MISR observations.

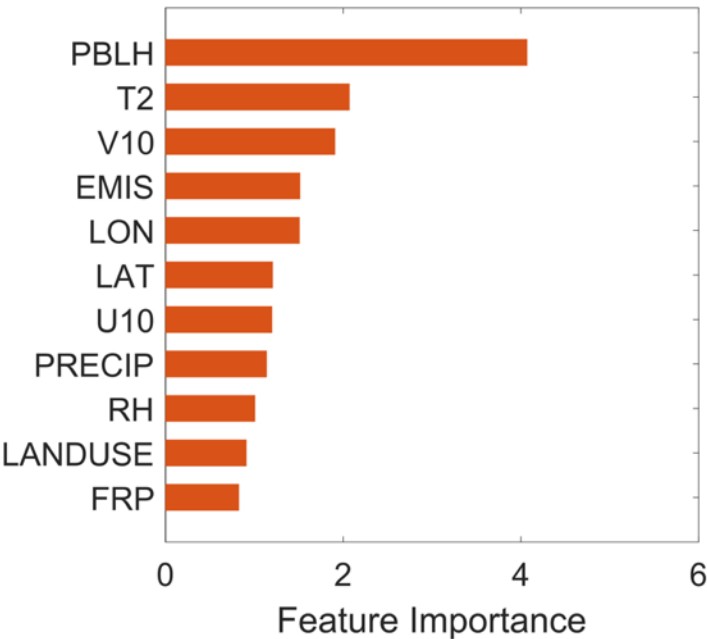

Figure 3. The importance of each predictor in the resulting random forest model. PBLH stands for the daily mean PBL height; T2, daily mean air temperature at 2 m; V10, daily mean meridional wind speed at 10 m; EMIS, daily mean OC fire emission flux; LON/LAT, the longitude and latitude of the plume source point; U10, zonal wind speed at 10 m, PRECIP: daily total precipitation, RH: daily mean relative humidity; LANDUSE, land use classification; and FRP, maximum fire radiative power within the plume.


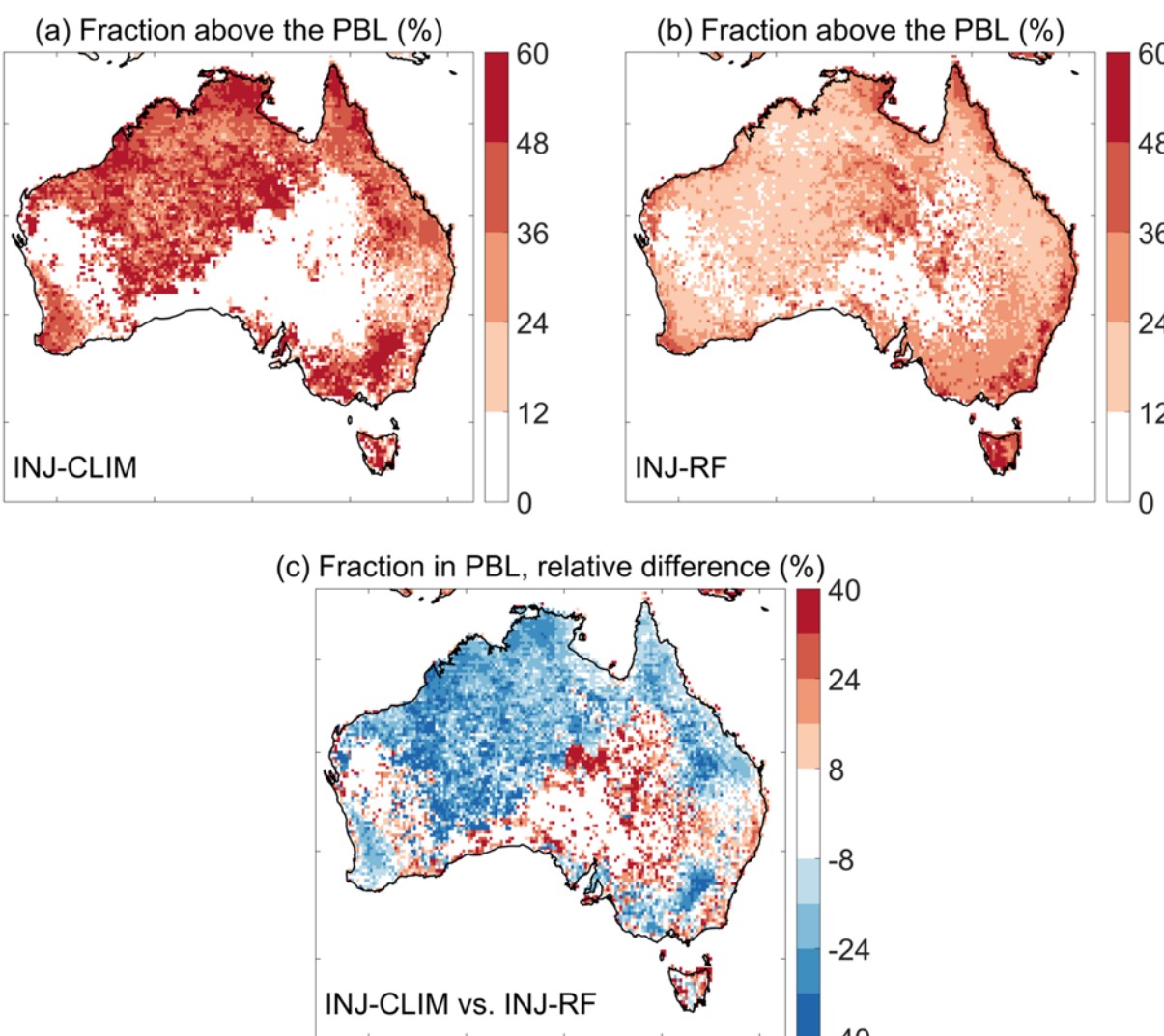

Figure 4. Spatial distributions of mean fractions of total OC fire emissions injected above the PBL (in units of %) estimated by (a) climatological method (INJ-CLIM) and (b) random forest method (INJ-RF), as well as (c) the percent differences in OC fire emissions within the PBL between the two methods relative to total OC fire emissions during April to the following January from 2009 to 2020.

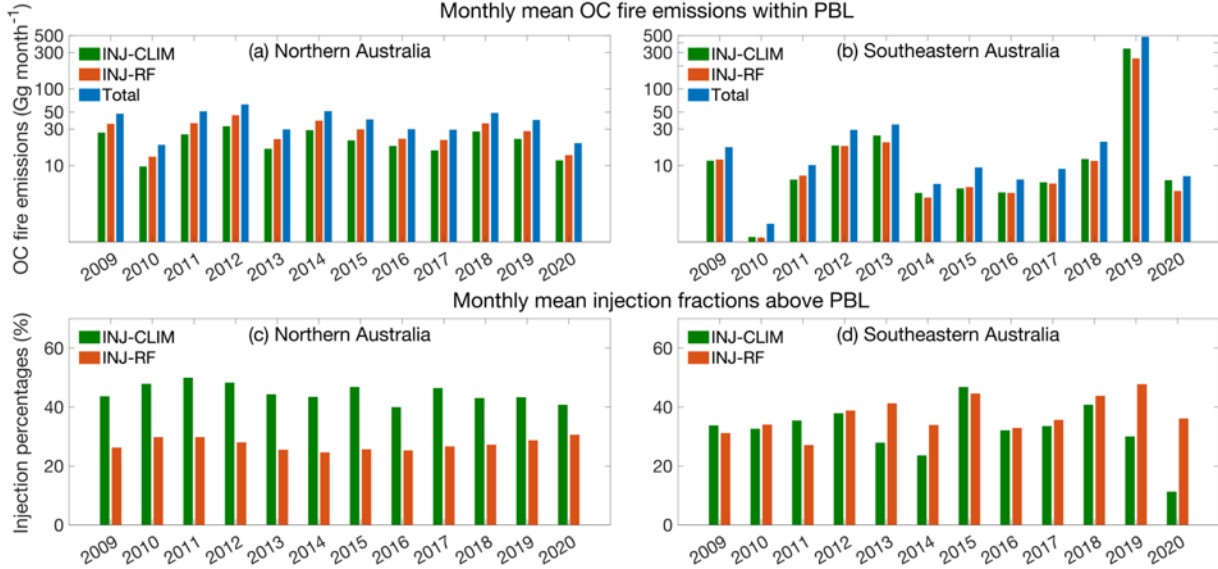


Figure 5. (a, b) Estimated monthly mean OC fire emissions released within the PBL and (c, d) annual mean fractions of OC fire emissions injected above the PBL based on the climatological method (INJ-CLIM, green bars) and the random forest method (INJ-RF, red bars) summed over all grid cells in northern Australia during April to December (left column) and in southeastern Australia during August to the following January (right column) from 2009 to 2020. Also shown are the monthly mean total fire emissions of OC during the respective fire seasons in northern and southeastern Australia (Total, blue bars in a and b). The y-axis of panel 5a and 5b are on a log scale. We assume that the plume injection fractions for BC fire emissions are the same as those for OC fire emissions.

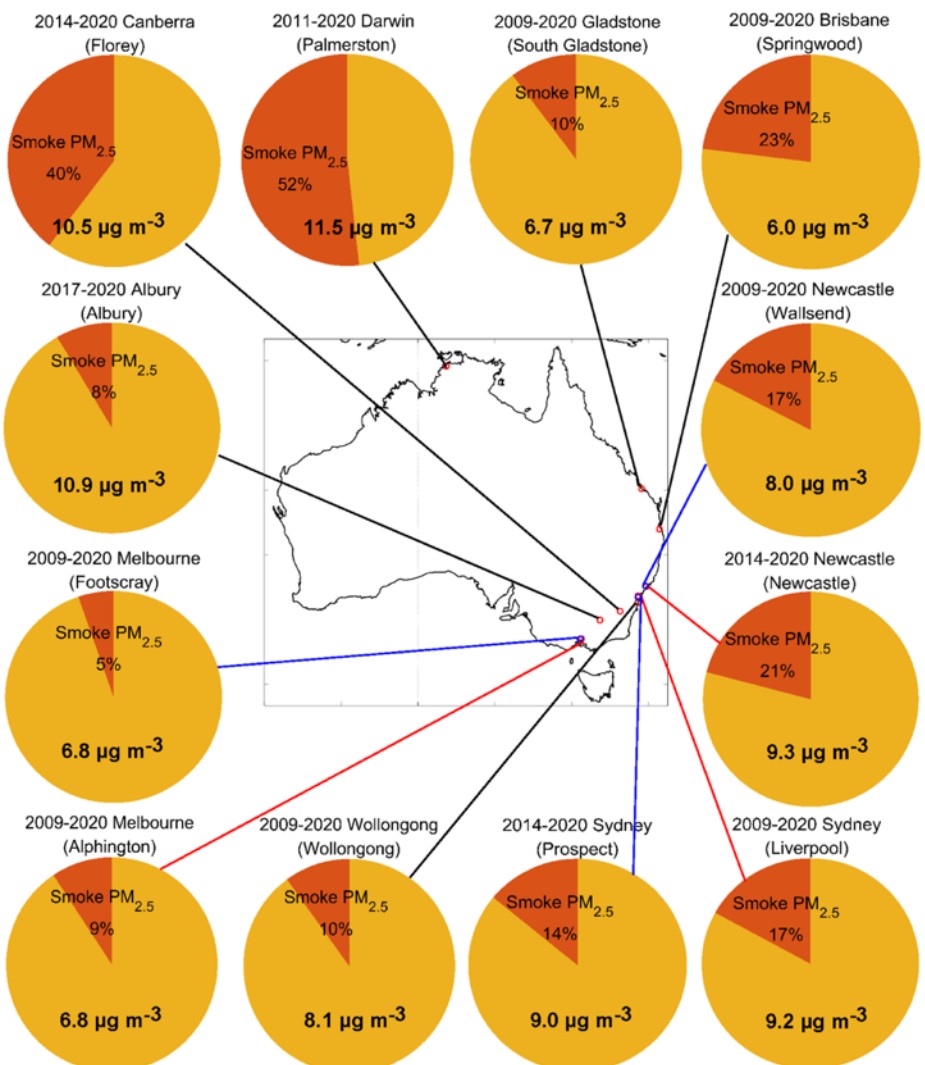

Figure 6. Contributions of simulated smoke PM$_{2.5}$ concentrations from the INJ-RF experiment to the observed total PM$_{2.5}$ concentrations (numbers on the pie charts) at 12 receptors averaged over the fire seasons of their respective observation periods (Table 3). Names of the observation sites are given in parentheses. Red sectors represent smoke contributions, while dark yellow sectors signify the differences between observed total PM$_{2.5}$ and simulated smoke PM$_{2.5}$ concentrations – i.e., the non-fire PM$_{2.5}$. Small circles on map represent the locations of these receptors. Different colors (red, blue, and black) are used to distinguish adjacent receptors.

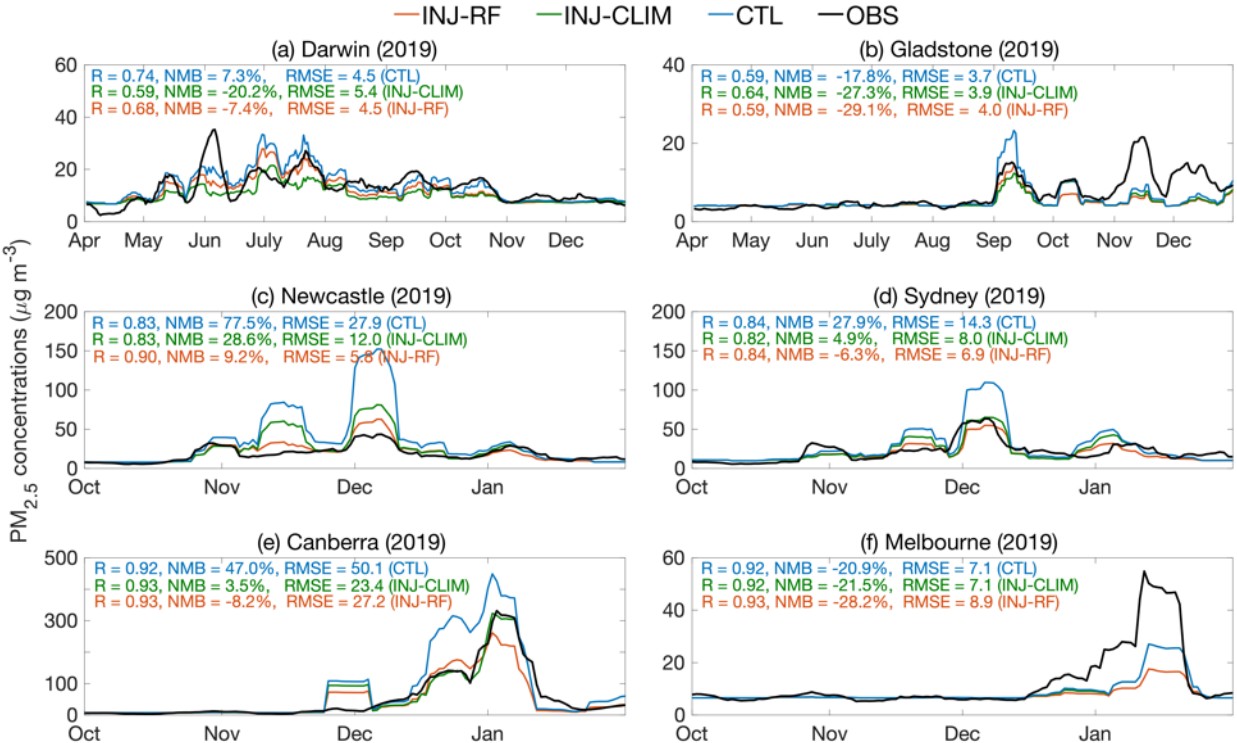

Figure 7. Time series of 10-day moving average of observed and simulated total PM₂.₅ concentrations from the CTL (blue), INJ-CLIM (green), and INJ-RF (red) experiments during the fire seasons of 2019-2020 at six sample receptors: (a) Darwin (Palmerston), (b) Gladstone (South Gladstone), (c) Newcastle (Wallsend), (d) Sydney (Liverpool), (e) Canberra (Florey), and (f) Melbourne (Footscray). Shown inset are the temporal correlation coefficients $R$, NMBs, and RMSEs of daily total PM₂.₅ concentrations compared to the surface measurements.

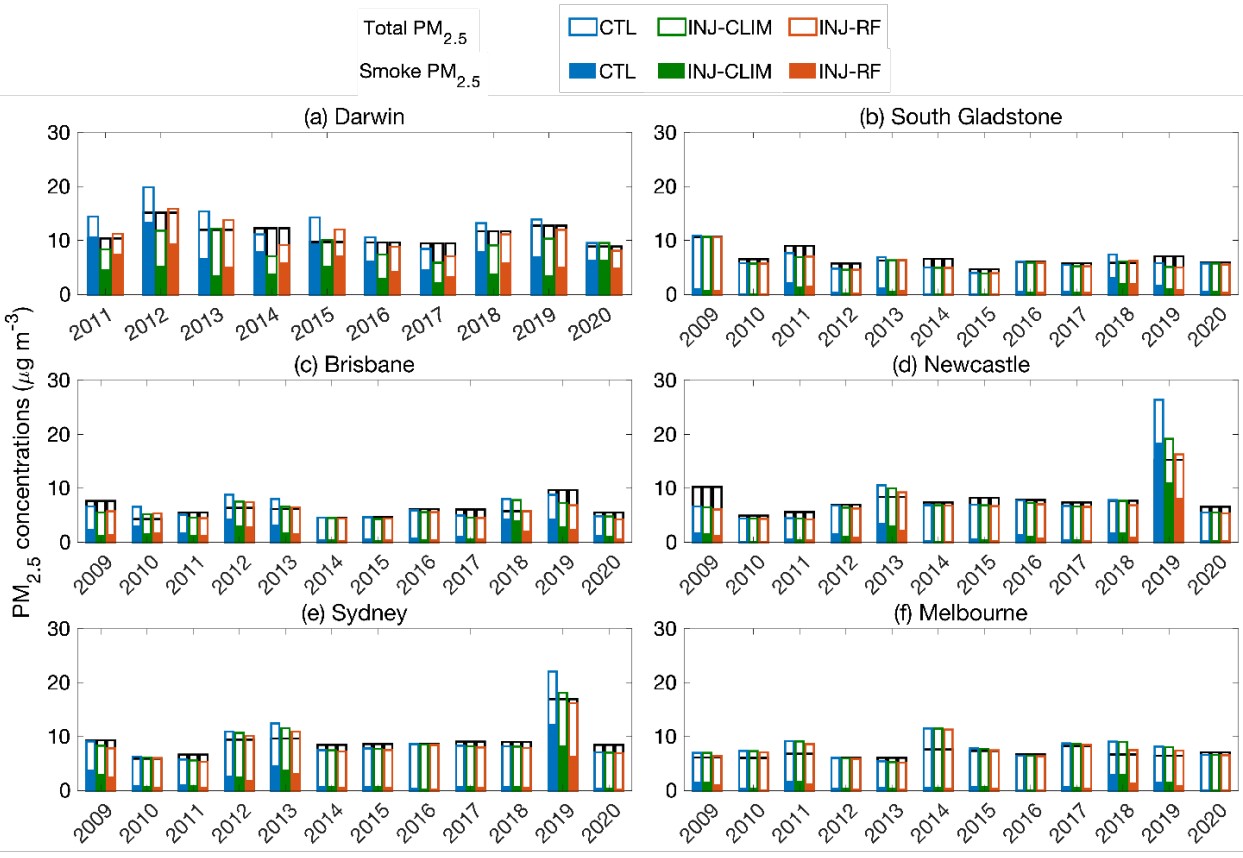

Figure 8. Mean simulated concentrations of smoke PM$_{2.5}$ and background PM$_{2.5}$ from the three sensitivity experiments (blue: CTL, green: INJ-CLIM, red: INJ-RF), as well as observed total PM$_{2.5}$ concentrations (black: OBS) in (a) Darwin (Palmerston), (b) Gladstone (South Gladstone), (c) Brisbane (Springwood), (d) Newcastle (Wallsend), (e) Sydney (Liverpool), and (f) Melbourne (Alphington). The different receptors have different observation periods. The modeled total PM$_{2.5}$ concentrations are designated by the height of the colored bars, consisting of smoke PM$_{2.5}$ (color-filled bars) and the background PM$_{2.5}$ (empty bars) in units of µg m$^{-3}$.

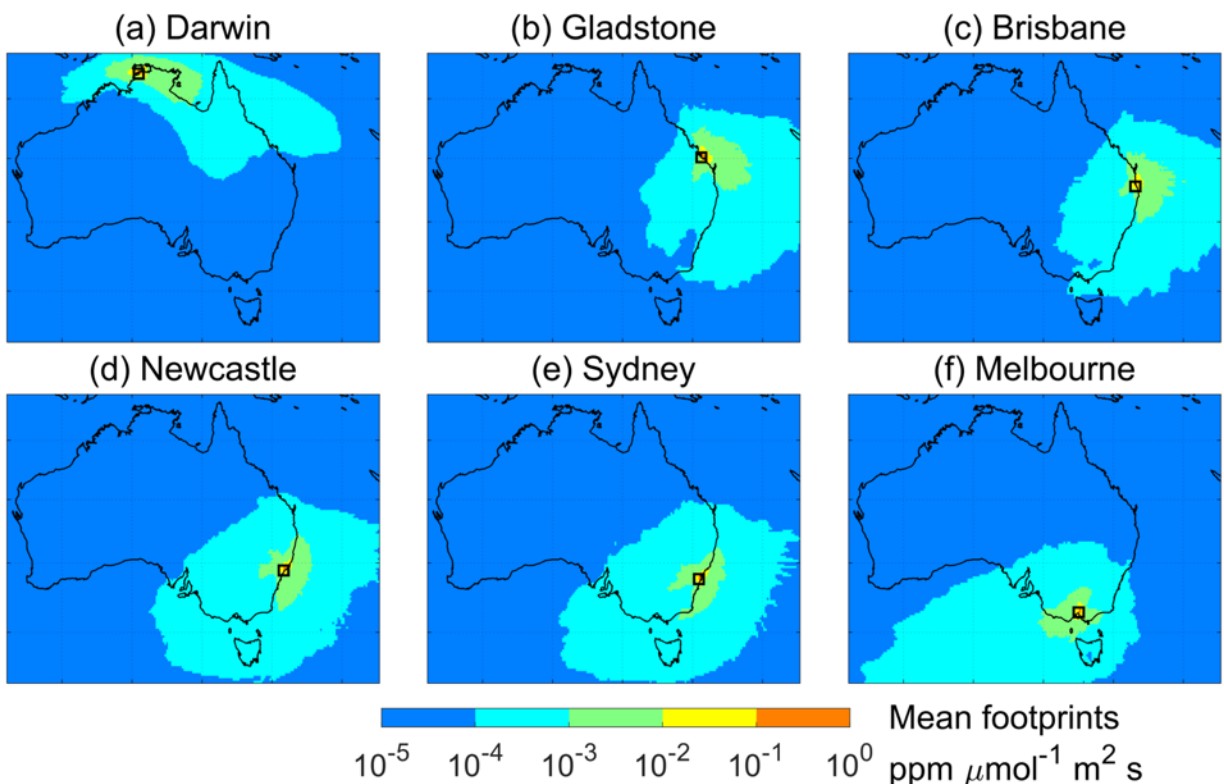

Figure 9. Mean of sensitivity footprints simulated by STILT in unit of ppm umol$^{-1}$ m$^2$ s during the fire seasons (April to December for Darwin and Gladstone; August to the following January for other cities) in (a) Darwin (Palmerston), (b) Gladstone (South Gladstone), (c) Brisbane (Springwood), (d) Newcastle (Wallsend), (e) Sydney (Liverpool) and (f) Melbourne (Alphington) from 2009 to 2020. The names in parentheses are site names. The black squares represent the locations of receptors.

1055

1060

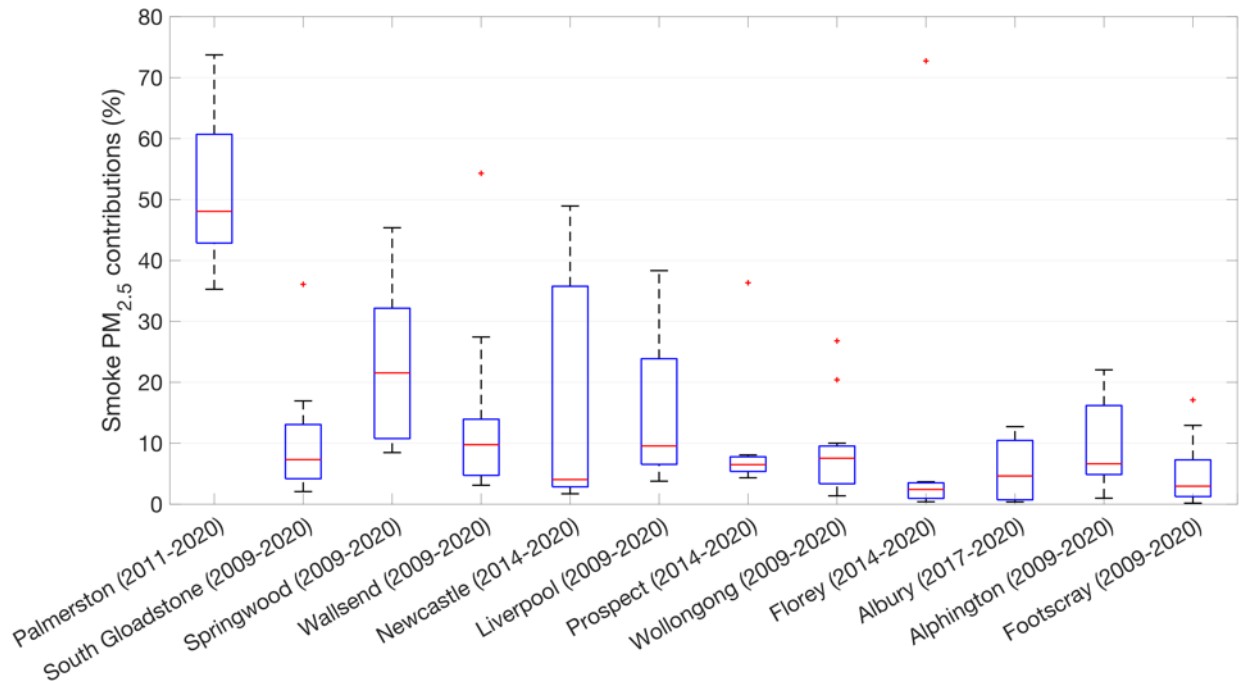

Figure 10. Boxplot of annual contributions of simulated smoke $PM_{2.5}$ concentrations from the INJ-RF experiment to observed total $PM_{2.5}$ concentrations at 12 receptors during the fire seasons of respective observations periods. The order of 12 receptors in this figure is based on the locations from north to south in Australia. The bottom, top, and red lines in the middle of each box are the 25[th] and 75[th] percentiles, as well as the median of all data. The distance between the 75[th] and 25[th] percentiles is the interquartile range. The lower and upper whisker limits represent the most extreme data values within 1.5 times the interquartile range. The data greater than 1.5 times outside the interquartile range are considered outliers and are shown as red crosses.