# Peer review of "Improved estimates of smoke exposure during Australia fire seasons: Importance of quantifying plume injection heights"

_EGUsphere, 2023_

## Author Comment (AC1)

**Response from authors**

Re: Review of Improved estimates of smoke exposure during Australia fire seasons: Importance of quantifying plume injection heights

October 20, 2023

We thank the reviewers for these helpful comments. We respond to each specific comment in details below. Referee comments are shown in red. Our replies are shown in black and modified text is shown in blue. The annotated page and line numbers refer to the revised copy of the manuscript.

**Referee comment 1**

The manuscript addresses the issue of accurately and efficiently calculating plume injection heights, aiming to enhance the accuracy of estimating smoke exposure during Australia's biomass-burning season. For this purpose, the authors employed two distinct methodologies to quantify the fractions of fire emissions injected above the planetary boundary layer: a climatological approach with the use of climatological monthly mean injection profiles from IS4FIRES and daily injection heights taken from the GFAS emission inventory and a novel machine-learning approach using random forest models trained using plume heights from MISR satellite instrument. The effect of wildfires on the daily smoke $PM_{2.5}$ concentrations in Australian cities is estimated using the STILT Lagrangian particle dispersion model.

The results presented in the manuscript reveal that the machine-learning approach generally outperformed the climatological in predicting the daily plume injection fractions. Furthermore, considering the injection fractions predicted by the random forest model leads generally improves the agreement between $PM_{2.5}$ concentrations (estimated by STILT) in Australian cities and surface measurements. Lastly, based on the findings from the Random Forest and STILT models, smoke-related $PM_{2.5}$ in Australian cities accounts for up to 52% of total $PM_{2.5}$ during the biomass-burning season.

Overall the manuscript is well-written, and the results are interesting. However, in my opinion, there are some issues to be addressed and some clarifications to be provided before its publication in ACP.

**Specific comments**

1. ll 61-62: You described the drawbacks of using MISR and CALIOP to estimate the plume injection heights. Are there any drawbacks regarding the use of TROPOMI?

   This is a good question. In comparison to the plume heights retrieved from MISR and CALIOP, the plume heights obtained from TROPOMI offer daily global coverage. However, TROPOMI data is available only from 2018 onwards, as the Copernicus Sentinel-5P satellite was launched in 2017. Therefore, we cannot utilize TROPOMI data for studying the long-term plume heights.

   **P4, Line 80-82, Introduction**

   The plume heights retrieved from TROPOMI offer daily global coverage, but TROPOMI data are available only from 2018 onwards and so cannot be utilized for long-term study.

2. Why did you choose PBL heights from MERRA-2? There are differences between the PBL heights provided by different datasets (Ding et al., 2021; Guo et al., 2021) How the uncertainties on the MERRA-2 PBL heights can affect your results?

   We chose the MERRA-2 to obtain the long-term reliable meteorological variables at high spatiotemporal resolution. MERRA-2 is the first long-term global reanalysis to assimilate space-based observations of aerosols and represent their interactions with other physical processes in the climate system. We also aim to make our tool useful for GEOS-Chem (Bey et al., 2001), a global chemical transport model that can be driven by meteorological data

from MERRA-2. The resulting plume injection fractions can be applied to distributing the fire emissions above the PBL in the GEOS-Chem.

**P6, Line 138-141, Section 2**

MERRA-2 is the first long-term global reanalysis to assimilate space-based observations of aerosols and represent their interactions with other physical processes in the climate system. This reanalysis is often used to drive chemical transport models such as GEOS-Chem (Bey et al., 2001; Keller et al., 2014; Kim et al., 2015).

**Reference**

Bey, I., Jacob, D. J., Yantosca, R. M., Logan, J. A., Field, B. D., Fiore, A. M., Li, Q., Liu, H. Y., Mickley, L. J., and Schultz, M. G.: Global modeling of tropospheric chemistry with assimilated meteorology: Model description and evaluation, Journal of Geophysical Research: Atmospheres, 106, 23 073–23 095, https://doi.org/10.1029/2001JD000807, 2001.

Keller, C. A., Long, M. S., Yantosca, R. M., Da Silva, A. M., Pawson, S., and Jacob, D. J.: HEMCO v1.0: a versatile, ESMF-compliant component for calculating emissions in atmospheric models, Geoscientific Model Development, 7, 1409–1417, https://doi.org/10.5194/gmd-7-1409-2014, 2014.

Kim, P. S., Jacob, D. J., Fisher, J. A., Travis, K., Yu, K., Zhu, L., Yantosca, R. M., Sulprizio, M. P., Jimenez, J. L., Campuzano-Jost, P., Froyd, K. D., Liao, J., Hair, J. W., Fenn, M. A., Butler, C. F., Wagner, N. L., Gordon, T. D., Welti, A., Wennberg, P. O., Crounse, J. D., St. Clair, J. M., Teng, A. P., Millet, D. B., Schwarz, J. P., Markovic, M. Z., and Perring, A. E.: Sources, seasonality, and trends of southeast US aerosol: an integrated analysis of surface, aircraft, and satellite observations with the GEOS-Chem chemical transport model, Atmospheric Chemistry and Physics, 15, 10 411–10 433, https://doi.org/10.5194/acp-15-10411-2015, 2015.

3. l 122: A brief definition of the "top and bottom heights of plumes", "MHMI", and "injection height" would be helpful for readers less familiar with these terms.

Done.

**P S2, Line 14-30, Appendix S1**

**Definition of the plume injection height parameters**

The Global Fire Assimilation System (GFAS, Rémy et al., 2017) relies on a 1-D plume rise model (Freitas et al., 2007, 2010) to calculate the top of the plume, the bottom of the plume, and the mean height of maximum injection (MHMI). This plume rise model is governed by equations based on the vertical motion, mass conservation, and the first thermodynamic law (Freitas et al., 2007). A schematic of this plume rise model is shown in Figure 1 of Rémy et al. (2017). The plume entrainment and detrainment profiles are estimated by functions of fire radiative power, fire area, ambient temperature, and wind profiles. The detrainment is used to define the MHMI, which is the average of plume heights where detrainment exceeds half of the maximum value. The heights of the top and bottom of the detrainment profile are defined as the top and bottom heights of the plume.

GFAS also relies on a semi-empirical parameterization IS4FIRES (Sofiev et al., 2012, 2013) to calculate the injection height, which is defined as the top of the plume. According to this algorithm (Eq. S1), the top of the plume ($H_p$) is a function of the PBL height ($H_{abl}$), fire radiative power (FRP), and Brunt-Vaisala frequency in the free troposphere ($N_{FT}$). The other coefficients are constant values.

$$H_p = \alpha H_{abl} + \beta \left(\frac{FRP}{P_{f0}}\right)^{\gamma} exp\left(\frac{-\delta N_{FT}^2}{N_0^2}\right) \qquad (S1)$$

4. Please provide a short discussion on the role of the plume injection height in surface air pollution based referring to new studies such as Li et al., 2023.

We have now added a brief discussion on this study in the Introduction section. Additionally, this study may explain the underestimate of surface PM$_{2.5}$ concentrations in some downwind regions in our study, as we assume that the injected fire emissions would stay above the PBL and would have no impact on surface PM$_{2.5}$ concentrations. In our study, the INJ-CLIM and INJ-RF methods estimate the plume injection fractions only in source grids, and they cannot estimate how much of fire emissions may get mixed downward in remote regions. Please see further discussion on this issue in Questions 6 and 16 (pages 6 and 14 of this document).

**P3, Line 62-68, Introduction**

The plume injection heights determine the vertical distribution of fire emissions, affecting surface smoke exposure, the long-range transport, and removal processes of emitted pollutants (e.g., Jian and Fu, 2014; Zhu et al., 2018). A recent study used three plume rise schemes in the Community Multiscale Air Quality model to study the plume injection heights and their impacts on air quality, indicating that higher plume injection heights led to lower surface $PM_{2.5}$ concentrations near the source region but higher concentrations in regions downwind due to the transport at higher altitudes followed by downward mixing (Li et al., 2023).

**Reference**

Li, Y., Tong, D., Ma, S., Freitas, S. R., Ahmadov, R., Sofiev, M., Zhang, X., Kondragunta, S., Kahn, R., Tang, Y., Baker, B., Campbell, P., Saylor, R., Grell, G., and Li, F.: Impacts of estimated plume rise on $PM_{2.5}$ exceedance prediction during extreme wildfire events: a comparison of three schemes (Briggs, Freitas, and Sofiev), Atmospheric Chemistry and Physics, 23, 3083–3101, https://doi.org/10.5194/acp-23-3083-2023, 2023.

5. l 260: "Smoke $PM_{2.5}$ is typically defined as the sum of the fire-related BC and organic matter (OM)" Please provide a reference. The non-biomass burning OM (e.g. biogenic organic aerosols) do not affect the concentrations?

Carbonaceous aerosols are the main components in wildfire smoke (Chow et al., 2011). There are some previous studies assuming that the smoke $PM_{2.5}$ is the sum of fire-related BC and fire-related OM (Koplitz et al., 2016, Cusworth et al., 2018, Li et al., 2020). We have added the citations in Section 2.3.1.

Smoke $PM_{2.5}$ concentrations are calculated via multiplying sensitivity footprints by fire emissions, which are affected by other emission sources. The non-biomass burning OM concentrations are included in non-fire $PM_{2.5}$ concentrations described in Section 2.4.

**P10, Line 271-275, Section 2.3.1**

The model yields the concentrations of fire-related black carbon (BC) and organic carbon (OC) particulate matter at each receptor within the source region via multiplying the sensitivity footprints by the fire emissions on daily timescales. Smoke $PM_{2.5}$ is typically defined as the sum of the fire-related BC and organic matter (OM) (Chow et al., 2011; Koplitz et al., 2016; Cusworth et al., 2018; Li et al., 2020).

**Reference**

Chow, J. C., Watson, J. G., Lowenthal, D. H., Antony Chen, L.-W., and Motallebi, N.: $PM_{2.5}$ source profiles for black and organic carbon emission inventories, Atmospheric Environment, 45, 5407–5414, https://doi.org/10.1016/j.atmosenv.2011.07.011, 2011.

Koplitz, S. N., Mickley, L. J., Marlier, M. E., Buonocore, J. J., Kim, P. S., Liu, T., Sulprizio, M. P., DeFries, R. S., Jacob, D. J., Schwartz, J., Pongsiri, M., and Myers, S. S.: Public health impacts of the severe haze in Equatorial Asia in September–October 2015: demonstration of a new framework for informing fire management strategies to reduce downwind smoke exposure, Environmental Research Letters, 11, 094023, https://doi.org/10.1088/1748-9326/11/9/094023, 2016.

Li, Y., Mickley, L. J., Liu, P., and Kaplan, J. O.: Trends and spatial shifts in lightning fires and smoke concentrations in response to 21st century climate over the national forests and parks of the western United States, Atmospheric Chemistry and Physics, 20, 8827–8838, https://doi.org/10.5194/acp-20-8827-2020, 2020.

6. ll 283-284: "We assume that the fire emissions injected above the PBL have no impact on the surface $PM_{2.5}$." Is it possible that this may introduce a low bias to the results?

We thank the reviewer for making this point. As discussed in Question 4 above, this assumption may cause an underestimate of simulated surface $PM_{2.5}$ concentrations in some downwind regions. For example, during the high-fire year of 2019, the modeled smoke $PM_{2.5}$ concentrations agree well with the observations in sites near the source regions (e.g., Sydney and Newcastle), but are underestimated in the downwind regions (e.g., Brisbane, Gladstone, and Melbourne). These low biases may be due to lack of the downward mixing of fire plumes from the high altitudes in downwind regions, which we are unable to take into consideration in this study. We clarify this in Section 2.3.2.

**P11, Line 301-304, Section 2.3.2**

However, the INJ-CLIM and INJ-RF methods estimate the plume injection fractions only in the source grids, and they are unable to estimate to what extent smoke plumes mix down to the surface in remote regions downwind. This assumption may lead to the low biases of surface smoke $PM_{2.5}$ in remote regions, which we discuss in Section 4.

7. Please add some references regarding the synoptic conditions favoring the occurrence of wildfires. i.e.: ll 324-325 (During this period, ... northern Australia), ll 326-327 (During the monsoon periods ... of wildfires) and ll 331-332 (when low-pressure systems ... coastal areas).

We have now added the references related to wildfire weather and Australian monsoon and have moved this paragraph to the Introduction section.

**P3, Line 42-51, Introduction**

The peak periods of wildfires over northern Australia are generally during the dry season (April to October), when the high-pressure systems located in southern Australia bring dry and warm southeasterly winds to the Top End and Far North Queensland (FNQ) (Australian Bureau of Meteorology, 2023b). The Australian monsoon also governs fire seasonality in northern Australia. During the monsoon periods from November to April, the prevailing winds shift to northwesterly, bringing moist air from the ocean and reducing the risk of wildfires (Australian Bureau of Meteorology, 2023a). In southern Australia, the burning season typically occurs in austral spring and summer (September to February) when fuels are abundant. However, fire activity in this region shows large interannual variability. The fire danger increases when low-pressure systems in Tasmania bring hot and dry westerly winds to the coastal areas (Australian Bureau of Meteorology, 2023b).

**Reference**

Australian Bureau of Meteorology: The Australian Monsoon,
http://www.bom.gov.au/climate/about/australian-climate-influences.shtml?bookmark=monsoon,
last access: 29 August 2023a.

Australian Bureau of Meteorology: Bushfire Weather, http://www.bom.gov.au/weather-services/fire-
weather-centre/bushfire-weather/index.shtml, last access: 29 August 2023b.

8. ll 345-348: According to Fig. 2, there are many cases where we observe an overestimation of injection fractions by INJ-CLIM compared to MISR observations, mainly in cases with relatively low fire emissions. Can you elaborate on this? Also, please show the normalized mean bias in Fig. 2 (and also Fig. 3, at least in Figs. 2b,3b)

Thank you for this question. We have added the discussion related to the overestimation of injection fractions by INJ-CLIM.

**P13, Line 353-358, Section 3.2**

We find that 90% of the overestimated injection fractions with relative low fire emissions are located in the north and central of Australia, a finding which we attribute to inaccuracies in the climatological plume profiles. The plume injection height of the plume profile is proportional to the PBL height in this method (appendix S1, Sofiev et al., 2013), and given the relatively deep PBL in this region, the injection fractions above the PBL tend to be overestimated.

We have combined the Figure 2a-2b and Figure 3a-3b as new Figure 2 and added the NMB in each subplot.

[Figure]

Figure 2. Scatter plots of plume injection fractions above the PBL calculated by (a) climatological method and by (c) random forest, compared to the MISR derived plume injection fractions. Scatter plots of mass fluxes of fire emissions injected above the PBL estimated by injection fractions from (b) climatological method and (d) random forest, compared to MISR observations, in units of kg m$^{-2}$ s$^{-1}$. Also shown are the $R^2$ values and the RMSEs and NMBs between the predictions and MISR observations.

9. ll 359-361: What about the low bias for large injection fractions? Please elaborate more on the potential causes of the biases.

There are two potential causes for the low bias for large injection fractions. We have elaborated them in Section 3.2.

**P13, Line 367-375, Section 3.2**

Our random forest model generally captures the plume injection fractions compared to the MISR observations, with a normalized mean bias (NMB) of 1.3% and a RMSE of 0.22. The model explains 53% of the variance in the injection fractions derived from MISR, with overestimates at the low end and underestimates at the high end of the distribution, which can be partly attributed to systematic biases associated with ensemble-tree machine learning regressions (Zhang and Lu, 2012; Belitz and Stackelberg, 2021). In addition, we include only 191 records of plume height retrievals in November 2019, most of which are associated with large injection fractions. This relatively limited plume record may not have been adequate to predict the plume behavior of intense fires with confidence.

**Reference**

Belitz, K. and Stackelberg, P.: Evaluation of six methods for correcting bias in estimates from ensemble tree machine learning regression models, Environmental Modelling Software, 139, 105006, https://doi.org/10.1016/j.envsoft.2021.105006, 2021.

Zhang, G. and Lu, Y.: Bias-corrected random forests in regression, Journal of Applied Statistics, 39, 151–160, https://doi.org/10.1080/02664763.2011.578621, 2012.

10. ll 364: How does the meridional wind speed at 10 m determine the atmospheric stability? I understand that this is probably true, but maybe it's better to add a short explanation or reference.

Wind speed at 10 m combined with other atmospheric factors is commonly used for the atmospheric stability classification (Mohan and Siddiqui, 1998). The atmospheric stability can be affected by advection. We have added the explanation and references in the manuscript.

**P14, Line 384-390, Section 3.3**

The first three variables are highly related to ambient atmospheric stability (Mohan and Siddiqui, 1998) and fire behavior (Schroeder and Buck, 1970). Wildfire smoke disperses more under higher PBL heights and unstable atmospheric conditions, which in turn may be

affected by the movement of warmer air into the area near the surface or colder air into the area aloft (Schroeder and Buck, 1970). Thermal advection can be highly related to the meridional wind speed. Fire emissions implicitly reflect both the fire intensity and fuel load. The combined effects of these factors thus influence the degree to which the smoke plume is injected above the boundary layer.

**Reference**

Mohan, M. and Siddiqui, T.: Analysis of various schemes for the estimation of atmospheric stability classification, Atmospheric Environment, 32, 3775–3781, https://doi.org/https://doi.org/10.1016/S1352-2310(98)00109-5, 1998.

Schroeder, M. J., and Buck, C. C.: Atmospheric stability, in: Fire Weather – A Guide for application of Meteorological Information to Forest Fire Control Operations, Agriculture Handbook 360, PMS 425-1., p. 49-67, U.S. Department of Agriculture Forest Service, Washington, D.C., 1970.

11. l 371: "incomplete knowledge of the local temperature profile" Can you elaborate more on this?

Previous studies have attempted to directly correlate plume height with the fire radiative power (FRP). However, the relationship between FRP observations and the convective heat flux driving the plume rise depends on atmospheric stability. Therefore, the weak correlation between observed FRP and plume heights may be due to the incomplete knowledge of local temperature profiles that determine atmospheric stability (Kahn et al., 2007). We have included this explanation in Section 3.3.

**P14, Line 393-397, Section 3.3**

This weak correlation can be traced in part to clouds or smoke obscuring fires from satellite detection or to incomplete knowledge of the local temperature profile. Previous studies have attempted to directly correlate plume injection heights with FRP observations. However, the relationship between observed FRP and the convective heat flux driving the plume rise depends in large part on the local temperature profile which may not be well known (Kahn et al., 2007).

12. sec 3.1 I believe that it would be both interesting and useful for the readers to provide some connection between the results presented in Fig. 4 and those discussed in the previous sections of the manuscript. E.g., I can see that over coastal SE Australia (a region where the fire emissions are very pronounced according to Fig. 1) the mean fractions of total OC fire emissions injected above the PBL are larger when calculated using the random forest method. Is this connected to the fact that RF capture better strong fire events compared to the climatological method?

We thank the reviewer for this comment. The random forest method better captures the severe fire events in southeastern Australia compared to the climatological method because it includes the daily fire emissions and FRP as predictors to estimate the plume injection fractions. The climatological method relies on the climatological plume profiles, which are unable to represent the extreme fire events and the large interannual variability. We have revised the manuscript to include this discussion.

**P14-15, Line 415-419, Section 3.4**

In southeastern Australia, we find that the spatial distribution of large plume injection fractions predicted by random forest is highly correlated with that of high OC fire emissions in coastal areas (Figure 1). Given the good match of these injection fractions with MISR observations, we conclude that our random forest model better captures extreme wildfire events compared to the climatological method due to inclusion of daily fire emissions and FRP as predictors.

13. ll 424-426 and 4.1, generally: Can you elaborate more on why there are many cases with no improvement when using the INJ-RF, based also on the results presented in Table S1? Also, you state (ll 444-446) that in Melbourne the model has problems in reproducing successfully the daily variations of $PM_{2.5}$ during low fire years. However even during 2019 (Fig. 7f) the performance of INJ-RF is not good in this city (it's worse than CTL and INJ-CLIM).

This a good point. We have included the discussion on the cases with little improvement when using the INJ-RF experiments in Section 4.1.

**P16, Line 458-466, Section 4.1**

However, there are large biases in simulated total $PM_{2.5}$ concentrations from the INJ-RF experiment compared to the observations in Gladstone, Brisbane, Wollongong, and Albury. In Gladstone, Wollongong, and Albury, we also find the low biases in simulated total $PM_{2.5}$ concentrations from the CTL experiment, indicating that the total fire emissions from the original GFED 4.1s or the estimated non-fire $PM_{2.5}$ concentrations may be underestimated. The inclusion of plume injection in the INJ-RF and INJ-CLIM experiments thus aggravate low biases in simulated smoke $PM_{2.5}$ concentrations over the three cities. In Brisbane, we speculate that these biases arise from neglect in our model setup from downward mixing of smoke plumes in remote regions (Section 2.3.2).

**P16-17, Line 480-484, Section 4.1**

In Melbourne, three experiments capture fire events from December to January with temporal correlation coefficients over 0.92. However, the simulated total $PM_{2.5}$ concentrations are underestimated with NMBs ranging from -28.2% to -20.9% in all three experiments. Again, the peak values of smoke $PM_{2.5}$ concentrations are also unable to be captured by CTL experiment, which may be attributed to the low biases from the fire emission inventory.

**P18, Line 527-532, Section 4.2**

In 2011 and 2019, however, INJ-RF underestimates total $PM_{2.5}$ by 22% and 29.5%. The significant underestimates of total $PM_{2.5}$ can be partially attributed to the low biases in the fire emission inventory, which also leads to 15% and 18% underestimates of total $PM_{2.5}$ from the CTL experiment. Another reason may be neglect in our model setup of downward mixing of smoke far from the source regions. During the fire seasons in 2011 and 2019, Gladstone experiences the impacts of smoke from both local and remote burning regions in eastern coastal area.

14. l 458: 2016 or 2015?

Fixed.

**P17, Line 508-510, Section 4.2**

In the CTL experiment, simulated total $PM_{2.5}$ is 16.7% higher than the observations on average, with overestimates increasing to 31%-47% during the years with stronger fire emissions (e.g., 2011, 2012, and 2015).

15. l 470: "2 to 36%" according to which method?

Thanks for your question. The 2 to 36% wildfire contributions to total $PM_{2.5}$ are based on the INJ-RF method, including the plume injection fractions predicted by random forest. We have clarified this in Section 4.2.

**P18, Line 521-522, Section 4.2**

We find that annual mean wildfire contributions to total $PM_{2.5}$ varies greatly at this site, from 2% to 36% over the last decade based on the results of INJ-RF experiment.

16. ll 489-495: In Brisbane I see that the INJ-CLIM method performs slightly better than INJ-RF even during years with relatively large fires (e.g. 2019). Generally, I can see a similar performance for INJ-CLIM and INJ-RF in this city. Also according to Table S1, the NMB is larger for INJ-RF.

We thank the reviewer for pointing this out. Please refer to the responses for Questions 4 and 6. We have added the discussion in Section 4.2.

**P18-19, Line 545-552, Section 4.2**

In contrast, the INJ-RF experiment best matches the smoke $PM_{2.5}$ simulations in the cities near the burning regions during these high-fire years. For example, INJ-RF and INJ-CLIM reduce the large CTL overestimate of total $PM_{2.5}$ concentrations in Newcastle from 73% to 6.6% (INJ-RF) and 25.5% (INJ-CLIM) during 2019. But in remote downwind regions, both INJ-RF and INJ-CLIM experiments underestimate the smoke $PM_{2.5}$ concentrations in 2019, probably due to neglect in our model of downward mixing of fire plumes from high altitudes. The INJ-CLIM experiment estimates more fire emissions remaining within the PBL, which yields a smaller low bias in Brisbane.

17. sect 4.3 The discussion on the contribution of long-term smoke $PM_{2.5}$ to regional air quality is very interesting (results shown in Fig. S4). Consider moving Fig. S4 to the main manuscript.

We thank the reviewer for this suggestion. We have moved the figure to the main manuscript as new Figure 10.

18. ll 571-573: The NMBs do not agree with those in Fig. 7.

The NMBs here (P21, Line 640-642) are calculated based on the results in Figure 8, which compare the mean values of total observed and simulated $PM_{2.5}$ concentrations during the fire seasons. The NMBs in Figure 7 are calculated by comparing the simulated and observed 10-day moving averages of total $PM_{2.5}$ concentrations during the fire seasons in 2019.

19. I agree that the performance of the RF model approach is generally very satisfactory (and in most cases yields better results compared to the climatological approach). However, I suggest that the authors moderate somewhat their expressions regarding the performance of this method.

The reviewer makes a valuable suggestion. We have revised the expressions regarding the performance of the INJ-RF method in the Discussion and conclusion section.

**P21, Line 629-631, Section 5**

The random forest model predicts plume behavior that best agrees with observed surface PM$_{2.5}$, especially over the receptors near the burning regions during most high-fire years.

**P21-22, Line 642-650, Section 5**

However, the two methods are unable to quantify the possible downward mixing of fire smoke plumes in downwind regions and the subsequent impact on surface air in these regions. The INJ-RF method leads to more pronounced underestimates of surface PM$_{2.5}$ concentrations in Brisbane, Gladstone, and Melbourne as it predicts relatively small fractions of fire emissions remaining within the PBL in the source grids compared to the INJ-CLIM method. In this study, fire emissions injected above the PBL are assumed to have no impacts on surface PM$_{2.5}$ concentrations downwind. Future studies can explore the impacts of the long-range transport and downward mixing of fire emissions on surface smoke concentrations by applying the estimated injection fractions to 3-D chemical transport models.

**Referee comment 2**

In this paper, the authors compare three different methods for estimating plume injection heights/emission fractions above and below PBL: (1) a control with all emissions in the boundary layer, (2) using climatological injection heights from MISR and (3) using machine learning to estimate the injection height. These injection heights are used with the STILT model to estimate and compare surface concentrations in different cities across Australia.

The paper is generally well-written, and the methods seem sound. However, I felt like the methods section was very, very long and then was followed by sparse actual results and a lacking discussion section. I was left with many questions about their results. In general, I would have liked a greater discussion of where there were biases and hypotheses about what might be the cause, along with more discussion about when and where their machine learning injection height estimates did not improve the comparisons (line 475-476 seems like something that could be further explored). The Discussion and Conclusion section is mainly just a rehashing of results rather than a synthesis, comparison with previous studies, and necessary future work (rather than just apply it elsewhere and use more data). The abstract mentions that their model has the best agreement in several key cities, but it was not the best in other cities and other time periods. I'm not overall convinced that it is better than using the climatological injection heights. Before it can be published, it needs to provide more insight and analysis of the results.

We thank the reviewer for these helpful comments and questions. In response, we have carefully revised the manuscript to (1) reduce the length of method section, (2) include detailed hypotheses and analysis on the biases of estimated injection fractions and the resulting smoke $PM_{2.5}$ concentrations in Sections 3 and 4, and (3) improve the Discussion and Conclusion section. Please refer to the revised manuscript with tracked changes for the above modification. We respond to each specific comment in details below.

**Specific comments**

1. Abstract, line 29-31: State what this 63% is in comparison to in the sentence.

We have clarified this issue in the Abstract.

**P1, Line 29-32, Abstract**

However, using climatological injection profiles cannot capture well the spatiotemporal variability of plume injection fractions, resulting in a 63% underestimate of daily fire emission fluxes injected above the PBL in comparison with those fluxes derived from MISR injection fractions.

2. Line 72: PRM is only used two more time in the manuscript, I'd just write plume rise model each time to cut down on abbreviations.

We have removed the PRM abbreviation in our manuscript.

**P4, Line 86-88, Introduction**

Both GFAS and IS4FIRES rely on a plume rise model (Freitas et al., 2007, 2010) and semi-empirical parameterization (Sofiev et al., 2012; 2013) to determine injection heights.

**P6, Line 144-147, Section 2.1**

These parameters are calculated with two distinct algorithms: the one-dimensional plume rise model (Freitas et al., 2007, 2010; Rémy et al., 2017) and the IS4FIRES parameterization (Sofiev et al., 2012; 2013). The plume rise model predicts the daily vertical velocity, horizontal plume velocity, temperature, and plume radius;

**P6, Line 161-163, Section 2.1**

In this study, we regrid all datasets to a common 0.25° × 0.25° resolution, and then compare the MHMI derived from the plume rise model with the associated daily mean PBL height

from MERRA-2 to determine whether the fire emission should be lifted above the PBL at each grid cell.

3. Line 76: MHMI needs more explanation.

We have included the definition and description of MHMI in Supplement Information.

**P6, Line 141-144, Section 2.1**

We use the daily injection heights compiled by the GFAS emission inventory (Rémy et al., 2017), which provides four parameters representing the vertical extent of each smoke plume at 0.1° × 0.1° resolution: the top and bottom heights of plumes, the MHMI, and injection height (described in appendix S1).

**PS2, Line 14-24, Appendix S1**

**Definition of the plume injection height parameters**

The Global Fire Assimilation System (GFAS, Rémy et al., 2017) relies on a 1-D plume rise model (Freitas et al., 2007, 2010) to calculate the top of the plume, the bottom of the plume, and the mean height of maximum injection (MHMI). This plume rise model is governed by equations based on the vertical motion, mass conservation, and the first thermodynamic law (Freitas et al., 2007). A schematic of this plume rise model is shown in Figure 1 of Rémy et al. (2017). The plume entrainment and detrainment profiles are estimated by functions of fire radiative power, fire area, ambient temperature, and wind profiles. The detrainment is used to define the MHMI, which is the average of plume heights where detrainment exceeds half of the maximum value. The heights of the top and bottom of the detrainment profile are defined as the top and bottom heights of the plume.

4. Line 90-92: But may not include chemistry, other PM sources, or background smoke.

We have added these drawbacks of Lagrangian modeling in the Introduction section.

**P5, Line 107-109, Introduction**

However, Lagranigian modeling usually lacks chemical processes and is unable to capture background $PM_{2.5}$ concentrations from other anthropogenic and natural sources.

5. Line 100: It also requires that meteorology, emissions, and the injection heights are correct.

We have again added this to the Introduction.

**P5, Line 118-119, Introduction**

Furthermore, the accuracy of simulated smoke $PM_{2.5}$ concentrations in these models depends on reliable meteorology, biomass burning emissions, and plume injection heights.

6. Line 105: remove "improved" as it hasn't been shown to be improved yet.

Done.

**P5, Line 123-124, Introduction**

We first focus on two methods to quantify the daily fraction of fire emissions within and above the PBL:

7. Line 130: remove "need to".

Done.

**P6, Line 152, Section 2.1**

In addition to plume height, we also determine the mass fraction of smoke emitted above the PBL.

8. Line 157-8: Remove "for a limited set of months" as this is discussed later with the actual months listed.

Done.

**P7, Line 178, Section 2.2.1**

The plume heights used for training are those observed by the MISR instrument.

9. Line 181: what is the spatial distribution of these MISR records over Australia?

This is a good question. We have added the Figure S1 in the Supplementary Information to show the spatial distribution of all MISR records over Australia. In general, the MISR records are distributed in the coastal areas of northern and southeastern Australia.

**P8, Line 198-199, Section 2.2.1**

These MISR plume records are mainly distributed over the coastal areas of northern and southern Australia (Figure S1).

[Figure]

Figure S1. Spatial distribution of all MISR plume records over Australia used in this study.

10. Line 283-284: They can't mix down?

The reviewer makes a good point. It is possible for injected fire plumes above the PBL to travel long distances and mix down to the surface in remote downwind regions. In our model setup, we assume that the injected fire emissions above the PBL have no impact on the surface $PM_{2.5}$. This assumption may lead to an underestimate of surface $PM_{2.5}$ concentrations in some downwind regions. We revised the manuscript in Section 2.3.2 to clarify this point.

**P11, Line 301-304, Section 2.3.2**

However, the INJ-CLIM and INJ-RF methods estimate the plume injection fractions only in the source grids, and they are unable to estimate to what extent smoke plumes mix down to the surface in remote regions downwind. This assumption may lead to the low biases of surface smoke $PM_{2.5}$ in remote regions, which we discuss in Section 4.

11. Line 321-335 seems like intro section information.

We have simplified Section 3.1 and moved some content to the Introduction section.

**P3, Line 42-51, Introduction**

The peak periods of wildfires over northern Australia are generally during the dry season (April to October), when the high-pressure systems located in southern Australia bring dry and warm southeasterly winds to the Top End and Far North Queensland (FNQ) (Australian Bureau of Meteorology, 2023b). The Australian monsoon also governs fire seasonality in northern Australia. During the monsoon periods from November to April, the prevailing winds shift to northwesterly, bringing moist air from the ocean and reducing the risk of wildfires (Australian Bureau of Meteorology, 2023a). In southern Australia, the burning season typically occurs in austral spring and summer (September to February) when fuels are abundant. However, fire activity in this region shows large interannual variability. The fire danger increases when low-pressure systems in Tasmania bring hot and dry westerly winds to the coastal areas (Australian Bureau of Meteorology, 2023b).

**P12, Line 339-345, Section 3.1**

In northern Australia, the two main burning regions are located in the Top End and FNQ, which are covered by eucalypt forests and woodlands. In southeastern Australia, burning regions are mainly distributed in coastal eucalypt forested areas in New South Wales and Victoria, as well as in the Australian Capital Territory. In this study, we focus on the smoke exposure during April to December in northern Australia and August to January of the next year in southeastern Australia. In 2020, fire activity in southeastern Australia continued to some extent into February, but this lengthening of the typical fire season was unusual (Ellis et al., 2022).

12. Line 363-373 Discussion of Figure 3c should come before the evaluations (before 3.2)

The reviewer's comment is helpful. We agree that the Figure 3c should be a separate figure (new Figure 3). The discussion of Figure 3c is now included in a new Section 3.3 titled "Predictor importance for predicting plume injection fractions by random forest." However, we would like to keep the order of these sections unchanged. This is because we discuss the climatological method first, and then the random forest method, following the same order as in the Methods and data section.

**P13-14, Line 380-399, Section 3.3**

**3.3 Predictor importance for predicting plume injection fractions by random forest**

Figure 3 shows the importance of each predictor from the random forest model, which is calculated as described in Section 2.2.3. Larger values indicate greater importance. We find that the important variables include daily mean PBL height (PBLH), air temperature at 2 m (T2), meridional wind speed at 10 m (V10), and the corresponding fire emissions (EMIS). The first three variables are highly related to ambient atmospheric stability (Mohan and Siddiqui, 1998) and the fire behavior (Schroeder and Buck, 1970). Wildfire smoke disperses more under higher PBL heights and unstable atmospheric conditions, which in turn may be affected by the movement of warmer air into the area near the surface or colder air into the area aloft (Schroeder and Buck, 1970). Thermal advection can be highly related to the

meridional wind speed. Fire emissions implicitly reflect both the fire intensity and fuel load. The combined effects of these factors thus influence the degree to which the smoke plume is injected above the boundary layer. The maximum FRP within the plume is relatively less important predicting injection fractions above the PBL, consistent with previous studies which documented the weak correlation between FRP and injection height (Kahn et al., 2007; Val Martin et al., 2012). This weak correlation can be traced in part to clouds or smoke obscuring fires from satellite detection or to incomplete knowledge of the local temperature profile. Previous studies have attempted to directly correlate plume injection heights with FRP observations. However, the relationship between observed FRP and the convective heat flux driving the plume rise depends on the atmospheric structure determined by local temperature profiles (Kahn et al., 2007). In addition, the satellite pixels may be only partly filled by fire, leading to an underestimate of the heat flux driving plume rise.

13. Line 426: change to "on modeling smoke concentration for exposure estimates in Australia."

Done.

**P16, Line 456-458, Section 4.1**

At most sites, the results from the INJ-RF and INJ-CLIM experiments yield relatively lower RMSEs and NMBs against observations compared to the CTL experiment, indicating the importance of considering plume injection heights on modeling smoke concentration for exposure estimates in Australia.

14. Line 429: I'm not really sure this is a good idea for discussing exposure. I would have also liked the statistics from the daily observations included in at least the supplement (rather than just the 10-day averages). I'm also not sure how long smoke events normally last in these different areas.

Thank you for pointing out this issue. We have clarified the reasons for using 10-day averages in Section 4.1. During the wildfire seasons in Australia, the wildfire events that significantly increase the total $PM_{2.5}$ concentrations usually last for days to weeks. In this study, we focus on the contributions of smoke $PM_{2.5}$ to synoptic-scale variation of total

$PM_{2.5}$. We also calculated the statistics of simulated and observed daily total $PM_{2.5}$ over each receptor's observing period and included them in Table S2.

**P16, Line 469-473, Section 4.1**

We use the 10-day averages of simulated total $PM_{2.5}$ concentrations to reduce the impacts of weather conditions on day-to-day variability of non-fire $PM_{2.5}$, which is set to a constant value for each year at each receptor in our study, and to smooth out the response of smoke $PM_{2.5}$ to modeled fluctuations in fire activity. These fluctuations depend on the daily scale factors provided by GFED 4.1s and are somewhat uncertain.

Table S2. Statistics for daily mean PM$_{2.5}$ concentrations simulated by CTL, INJ-CLIM, and INJ-RF experiments, compared to the ground-based observations at 12 receptors. The daily mean concentrations are calculated over each receptor's observing period. Shown are the temporal correlation coefficients $R$, NMBs, and RMSEs of daily total PM$_{2.5}$ concentrations compared to the surface measurements.

| Cities (site) | Observation periods (Locations) | R | | | NMB | | | RMSE (µg m$^{-3}$) | | |
|---|---|---|---|---|---|---|---|---|---|---|
| | | *CTL* | *INJ-CLIM* | *INJ-RF* | *CTL* | *INJ-CLIM* | *INJ-RF* | *CTL* | *INJ-CLIM* | *INJ-RF* |
| **Darwin** (Palmerston) | 2011-2020 | 0.76 | 0.76 | 0.76 | 17.1% | -17.8% | -2.1% | 5.9 | 3.9 | 4.0 |
| **Gladstone** (South Gladstone) | 2009-2020 | 0.57 | 0.55 | 0.55 | -5.4% | -11.0% | -11.2% | 1.9 | 1.7 | 1.7 |
| **Brisbane** (Springwood) | 2009-2020 | 0.24 | 0.17 | 0.40 | 13.2% | 2.2% | -4.8% | 2.2 | 1.9 | 1.2 |
| **Newcastle** (Wallsend) | 2009-2020 | 0.53 | 0.48 | 0.52 | 16.1% | 2.3% | -6.0% | 6.6 | 3.9 | 2.4 |
| **Sydney** (Liverpool) | 2009-2020 | 0.40 | 0.38 | 0.37 | 14.8% | 6.8% | 0.4% | 4.8 | 3.4 | 2.8 |
| **Wollongong** (Wollongong) | 2009-2020 | 0.27 | 0.28 | 0.27 | -6.3% | -11.6% | -14.7% | 1.5 | 1.4 | 1.6 |
| **Melbourne** (Footscray) | 2009-2020 | 0.25 | 0.25 | 0.25 | -9.5% | -10.1% | -14.6% | 4.6 | 4.6 | 3.6 |
| **Melbourne** (Alphington) | 2009-2020 | 0.41 | 0.39 | 0.40 | 23.7% | 22.9% | 14.4% | 2.0 | 2.0 | 1.2 |
| **Albury** (Albury) | 2017-2020 | 0.93 | 0.93 | 0.93 | -22.2% | -23.7% | -31.7% | 5.6 | 5.8 | 7.3 |
| **Canberra** (Florey) | 2014-2020 | 0.67 | 0.63 | 0.68 | 19.3% | -8.6% | -16.0% | 22.5 | 17.7 | 15.3 |
| **Sydney** (Prospect) | 2014-2020 | 0.72 | 0.69 | 0.71 | 7.5% | -1.2% | -7.2% | 2.2 | 1.5 | 1.4 |
| **Newcastle** (Newcastle) | 2014-2020 | 0.59 | 0.50 | 0.55 | 28.9% | 9.2% | -1.6% | 6.9 | 4.0 | 2.7 |

15. Line 443: "the 10-day moving average of daily PM$_{2.5}$"

We have now included the statistics of simulated and observed daily total PM$_{2.5}$ concentrations over each receptor's observing period in Table S2.

**P17, Line 486-492, Section 4.1**

Table S2 shows the statistics for daily mean PM$_{2.5}$ concentrations simulated by CTL, INJ-CLIM, and INJ-RF experiments, compared to the ground-based observations at 12 receptors. These average concentrations reveal the long-term smoke exposure at these receptors. The three model experiments successfully reproduce the time series of daily PM$_{2.5}$ at most receptor cities except for Wollongong and Melbourne, with temporal correlation coefficients ranging from 0.4 to 0.93. In Wollongong and Melbourne (Footscray), where $R$=0.27 and 0.25, smoke PM$_{2.5}$ contributes only 10% and 5% of total PM$_{2.5}$ from 2009 to 2020 (Figure 6).

16. Table 1. Put in the shorthand names used in Figure 3 Feature Importance Plot

We have added the short name of predictors and target variable in Table 1.

Table 1. Predictors and target variable for the random forest model in this study

**Target variable**

| Short name | Description (unit) | Data source |
|---|---|---|
| Plume injection fraction | Daily plume injection fractions above the PBL (%) [a] | MISR Plume Height Project 2 (1.1 km for blue band, 275 m for red band; daily); MERRA-2 (0.5° latitude × 0.625° longitude, 1-hour) |

**Predictors**

| Short name | Description (unit) | Data source (spatial & temporal resolution) |
|---|---|---|
| LANDUSE | Land use classification (unitless) | MODIS Land Cover Climate Modeling Grid Version 6 (MCD12C1) (0.05° latitude × 0.05° longitude, yearly) |
| PBLH | Daily mean PBL height (m) | MERRA-2 (0.5° latitude × 0.625° longitude, 1-hour) |
| T2 | Daily mean air temperature at 2 m (K) | MERRA-2 (0.5° latitude × 0.625° longitude, 1-hour) |
| RH | Daily mean surface relative humidity (%) | MERRA-2 (0.5° latitude × 0.625° longitude, 3-hour) |
| U10 | Daily mean U-wind at 10 m (m s$^{-1}$) | MERRA-2 (0.5° latitude × 0.625° longitude, 1-hour) |
| V10 | Daily mean V-wind at 10 m (m s$^{-1}$) | MERRA-2 (0.5° latitude × 0.625° longitude, 1-hour) |
| PRECIP | Daily total precipitation (kg m$^{-2}$ s$^{-1}$) | MERRA-2 (0.5° latitude × 0.625° longitude, 1-hour) |
| EMIS | Daily mean OC biomass burning emissions (kg m$^{-2}$ s$^{-1}$) | GFED 4.1s (0.25° latitude × 0.25° longitude, daily) |
| LON | Longitude of the biomass burning emission grid cell (degree) | GFED 4.1s (0.25° latitude × 0.25° longitude, daily) |
| LAT | Latitude of the biomass burning emission grid cell (degree) | GFED 4.1s (0.25° latitude × 0.25° longitude, daily) |
| FRP | Maximum fire radiative power within the plume (MW) | MODIS/Terra Thermal Anomalies/Fire 5-Min L2 Swath 1km V061 (MOD14) (2030 km along swath × 2300 km across swath, 5-minute) |

[a] Fraction of plume pixels injected above the PBL within plume perimeter. Detailed calculation is described in Eq. (1) Section 2.2.1.

17. Figure 3a and 3b should be combined into Figure 2 and Figure 3c should just be its own plot as it seems out of place here.

We thank the reviewer for this suggestion. We have combined Figure 2a-2b and Figure 3a-3b into the new Figure 2. Figure 3c is new Figure 3. We have also combined Sections 3.2 and 3.3 on the evaluation of plume injection fractions calculated by these two methods into a new Section 3.2 titled "Evaluation of plume injection fractions calculated by climatological injection profiles and predicted by random forest".

[Figure]

Figure 2. Scatter plots of plume injection fractions above the PBL calculated by (a) climatological method and by (c) random forest, compared to the MISR derived plume injection fractions. Scatter plots of mass fluxes of fire emissions injected above the PBL estimated by injection fractions from (b) climatological method and (d) random forest, compared to MISR observations, in units of kg m$^{-2}$ s$^{-1}$. Also shown are the $R^2$ values and the RMSEs and NMBs between the predictions and MISR observations.

[Figure]

Figure 3. The importance of each predictor in the resulting random forest model. PBLH stands for the daily mean PBL height; T2, daily mean air temperature at 2 m; V10, daily mean meridional wind speed at 10 m; EMIS, daily mean OC fire emission flux; LON/LAT, the longitude and latitude of the plume source point; U10, zonal wind speed at 10 m, PRECIP: daily total precipitation, RH: daily mean relative humidity; LANDUSE, land use classification; and FRP, maximum fire radiative power within the plume.

18. Figure 5 Change to "and southeastern Australia (Total, blue bars in a and b) The y-axis feels confusing for a and b with one in linear and one in log scale. In b, there needs to be more labels on the y-axis. There seems to be lots more year-to-year variability in the INJ-CLIM for southeastern Australia compared to Northern Australia.

We have changed the caption as the reviewer suggests. In Figure 5b, we used a log scale for the y-axis because the range of annual fire emissions in southeastern Australia spans about two orders of magnitude. To improve the comparison, we have used the log scale for the y-axis and increased the label number in both Figure 5a and 5b.

The wildfire seasons in northern Australia are more consistent year-to-year compared to those in southeastern Australia. The interannual variabilities of plume injection fractions estimated by INJ-CLIM and INJ-RF methods both vary more dramatically in southeastern

Australia as the fire intensity and emissions change a lot in different years. We have clarified this in Section 3.4.

**P15, Line 425-431, Section 3.4**

Although there is large interannual variation of monthly mean total OC fire emissions, ranging from 18.6 Gg month$^{-1}$ to 62.9 Gg month$^{-1}$, neither method shows a long-term trend of plume injection fractions in northern Australia over the last decade. In southeastern Australia, the interannual changes in both fire emissions and plume injection fractions estimated by INJ-CLIM and INJ-RF methods are more pronounced from 2009 to 2020 compared to those in northern Australia. This is due to the dramatic changes in total amounts of wildfires and fire intensity in this region.

[Figure]

Figure 5. (a, b) Estimated monthly mean OC fire emissions released within the PBL and (c, d) annual mean fractions of OC fire emissions injected above the PBL based on the climatological method (INJ-CLIM, green bars) and the random forest method (INJ-RF, red bars) summed over all grid cells in northern Australia during April to December (left column) and in southeastern Australia during August to the following January (right column) from 2009 to 2020. Also shown are the monthly mean total fire emissions of OC during the respective fire seasons in northern and southeastern Australia (Total, blue bars in a and b). The y-axis of panel 5a and 5b are on a log scale. We assume that the plume injection fractions for BC fire emissions are the same as those for OC fire emissions.

19. Figure 6 (and Figure S5) isn't really useful. The pie charts don't provide much information and aren't in line geographically, so I'd either just put the numbers on the map, in a table, or just use Figure S4 as it is a more useful/less confusing plot with similar information. If you are attached to the pie charts, then they should represent the total $PM_{2.5}$ concentrations and be closer to the actual location.

Thank you for pointing out this issue. We have put the numbers of total $PM_{2.5}$ concentrations on the pie charts of Figure 6 and new Figure S4. We have moved the original Figure S4 to the main text as new Figure 10. As the receptors are concentrated in southeastern Australia, it's difficult to put the pie charts close to the actual locations. We do our best to arrange the pie charts and use the straight lines and circles to represent the locations.

[Figure]

Figure 6. Contributions of simulated smoke $PM_{2.5}$ concentrations from the INJ-RF experiment to the observed total $PM_{2.5}$ concentrations (numbers on the pie charts) at 12 receptors averaged over the fire seasons of their respective observation periods (Table S1). Names of the observation sites are given in parentheses. Red sectors represent smoke contributions, while dark yellow sectors signify the differences between observed total $PM_{2.5}$ and simulated smoke $PM_{2.5}$ concentrations – i.e., the non-fire $PM_{2.5}$. Small circles on map represent the locations of these receptors. Different colors (red, blue, and black) are used to distinguish adjacent receptors.

[Figure]

Figure S4. Contributions of simulated smoke PM$_{2.5}$ concentrations from the INJ-RF experiment to the observed total PM$_{2.5}$ concentrations (numbers on the pie charts) at 12 receptors averaged over the fire seasons of 2019. Names of the observation sites are given in parentheses. Red sectors represent smoke contributions, while dark yellow sectors signify the differences between observed total PM$_{2.5}$ and simulated smoke PM$_{2.5}$ concentrations – i.e., the non-fire PM$_{2.5}$. Small circles on map represent the locations of these receptors. Different colors (red, blue, and black) are used to distinguish adjacent receptors.

20. Figure 7. What are the statistics for the same day as opposed to the 10-day average? Do these smoke events often last a week or longer or are the peaks on shorter timescales?

We have now included the statistics of simulated and observed daily total PM$_{2.5}$ over each receptor's observing period in Table S2. During the wildfire seasons in Australia, the wildfire events that significantly increase the total PM$_{2.5}$ concentrations usually last for days to weeks. Please see the responses for Questions 14 and 15 for more detail.

21. Figure 9. The color bar levels are odd here. I can really only see two levels. Also, these suggest that for all of these locations, most of the air reaching the site comes from over the ocean. There's not discussion of how that might be impacting the results. Does this pattern look different on just fire/smoke days compared to the rest of the season?

We have updated the color bar to make the plot clearer and have included new discussion on Figure 9. Please also see response to Comment 23.

**P17, Line 500-505, Section 4.2**

Figure 9 shows the mean sensitivity footprints at six representative cities during the fire seasons from 2009 to 2020. The panels indicate the time-average source regions of the air masses reaching these receptors within 120 hours. When these air masses originate from burning regions, the surface PM$_{2.5}$ concentrations at the receptors show enhancements of smoke PM$_{2.5}$. In contrast, the impacts of wildfire smoke are quite small when the upwind source regions are over the ocean or non-burning areas.

[Figure]

Figure 9. Mean sensitivity footprints simulated by STILT in unit of ppm umol$^{-1}$ m$^2$ s during the fire seasons (April to December for Darwin and Gladstone; August to the following January for other cities) in (a) Darwin (Palmerston), (b) Gladstone (South Gladstone), (c) Brisbane (Springwood), (d) Newcastle (Wallsend), (e) Sydney (Liverpool) and (f) Melbourne (Alphington) from 2009 to 2020. The names in parentheses are site names. The black squares represent the locations of receptors.

22. Figures S1 and S2 should be combined.

We have combined Figures S1 and S2 as the new Figure S2.

[Figure]

Figure S2. Time series of 10-day moving average of observed and simulated total PM$_{2.5}$ concentrations from the CTL (blue), INJ-CLIM (green), and INJ-RF (red) experiments during the fire seasons at representative receptors in 12 receptors: (a) Darwin (Palmerston), (b) Gladstone (South Gladstone), (c) Newcastle (Wallsend), (d) Sydney (Liverpool), (e) Canberra (Florey), (f) Melbourne (Footscray), (g) Brisbane (Springwood), (h) Newcastle (Newcastle), (i) Sydney (Prospect), (j) Wollongong (Wollongong), (k) Albury (Albury), and (l) Melbourne (Alphington). Given in parentheses are the names of the observation

sites. The 10-day moving averages are calculated over each receptor's observing period, as indicated above the panels. Shown inset are the temporal correlation coefficients $R$, NMBs, and RMSEs of daily total $PM_{2.5}$ concentrations compared to the surface measurements.

23. I find Figure S6 really interesting and would like more discussion about it. These fire seasons are mentioned in the introduction, but perhaps some more discussion about how the different fire seasons might impact the model performance would be useful.

We thank the reviewer for this comment. We have added more discussion on the seasonality of fire seasons and the related model performance of STILT.

**P20, Line 588-598, Section 4.3**

Figure S5 shows the monthly mean contributions of smoke $PM_{2.5}$ at six representative sites over the last decade. In Darwin, mean smoke $PM_{2.5}$ contributions rise to over 50% from May to August, but fall to less than 20% from November to December. This seasonality is consistent over the last decade and can be traced to the influence of the Australian monsoon, as described in the Introduction. The wildfires in the Top End and FNQ usually last from April to December. From April to August, a high-pressure system is typically located in southern Australia. Southeasterly winds from this area are warm and dry, bringing smoke from burning regions in the Top End to Darwin. After September, the monsoon carries warm and moist oceanic air into Darwin from the northwest, limiting the impact of wildfire smoke emitted over the Top End and FNQ on air quality into the city. The STILT model usually yields a better performance capturing the patterns of sensitivity footprints due to the reliable meteorological variables provided by GDAS and GFS.

**P20, Line 606-610, Section 4.3**

However, the wildfire events in southeastern Australia experience large interannual variability, resulting in variable spatiotemporal distributions of fire emissions during fire seasons over the last decade. Air quality in the other five cities of southeastern Australia that we examine are affected by surface air fluxes from both land and ocean. The day-to-day variability of sensitivity footprints in these receptors are pronounced, which may be challenging for the STILT model to accurately reproduce.

---

## Author Response (AR2)

**Response from authors**

Re: Review of Improved estimates of smoke exposure during Australia fire seasons: Importance of quantifying plume injection heights

January 20, 2024

**Referee comment 1**

**General comment:**

Most of my remarks have been properly addressed by the authors, and, in my opinion, the manuscript has improved.
The discussion on the issues arising from assuming that fire emissions injected above the Planetary Boundary Layer (PBL) do not affect surface smoke concentrations is essential. Addressing this issue must be a priority in future work.

We thank the reviewer for these helpful comments. We respond to each specific comment in details below. The referee comments are shown in red. Our replies are shown in black and modified text is shown in blue. The annotated page and line numbers refer to the revised copy of the manuscript.

**Specific comments**

1. ll 362-363: "The large bias is mainly due to the underestimates of .. such as those in 2019."
I recommend adding a sentence here discussing the high model bias for small fire emissions. However, I acknowledge that they may not significantly contribute to the overall (negative) bias.

   We have discussed the possible reason for overestimated injection fractions (see details in Line 357-361) in northern and central Australia, which also leads to the high biases in injected emission fluxes with small fire emissions. We have added a sentence to clarify this.

**P13, Line 374-376, Section 3.2**

For low fire emissions, the climatological method shows high biases in injected emission fluxes above the PBL due to inaccurate climatological plume profiles in north and central of Australia.

2. ll 458-460: "However, there are also large bias ... Albury"
According to Table S1 the biases are also large in Newcastle and Canberra.

We have now implemented the discussion on the low biases in Newcastle (Wallsend) and Canberra. We also clarify this issue in Section 5.

**P16, Line 470-472, Section 4.1**

However, there are large biases in simulated total $PM_{2.5}$ concentrations from the INJ-RF experiment compared to the observations in Gladstone, Brisbane, Wollongong, Canberra, Newcastle (Wallsend), and Albury.

**P16, Line 477-479, Section 4.1**

In Brisbane, Canberra, and Newcastle (Wallsend), the low biases are relatively significant during the high-fire years of 2009 and 2019. We speculate that these biases arise from neglect in our model setup from downward mixing of smoke plumes in remote regions (Section 2.3.2).

**P22, Line 659-667, Section 5**

In Sydney and Newcastle, these two methods generate surface concentrations in better agreement with observations than the control simulation, with NMBs of -4.5% (INJ-RF) to -7.0% (INJ-CLIM) in Sydney and 6.6% (INJ-RF) to 25.5% (INJ-CLIM) in Newcastle.

However, neither method can quantify the possible downward mixing of fire smoke plumes in downwind regions and the subsequent impact on surface air, a process which may be especially important during more severe fire seasons when intense heat lofts greater quantities of smoke above the PBL in source regions. In Brisbane, Gladstone, and Melbourne, the INJ-RF method leads to more pronounced underestimates of surface $PM_{2.5}$ concentrations compared to INJ-CLIM, perhaps because of this shortcoming.

3. ll 478-480: "In particular ... November to January." The agreement is less evident in Sydney, as indicated by comparable statistical metrics between INJ-CLIM and INJ-RF in Figure 7.

The statistical metrics in Figure 7 are based on the whole fire seasons from August to January in the following year, which yield a smaller RMSE and a higher correlation coefficient in Sydney. During the peak time (November to mid-December) of the fire season in 2019, the NMBs for INJ-CLIM and INJ-RF are 12% and -5.0% in Sydney. In Newcastle, the NMBs for INJ-CLIM and INJ-RF are 77% and 31%. We have clarified this in Section 4.1

**P17, Line 491-494, Section 4.1**

In particular, compared to results from the INJ-CLIM experiment, the peak values of total $PM_{2.5}$ simulated by INJ-RF experiment agree best with observations in Newcastle and Sydney during the megafires of November to Mid-December, with the lowest NMBs of 31% and -5.0%.

4. Table S2: I agree that the three model experiments successfully reproduce the time series of daily $PM_{2.5}$ at most receptor cities, which is encouraging.
   However, I am concerned about the lack of a clear improvement in INJ-RF compared to INJ-CLIM in many receptors. A brief explanation or discussion about this point would be helpful.

Thank you for this suggestion. We agree that the improvement in INJ-RF compared to INJ-CLIM are not significant in some receptors during the fire seasons over the last decade. We have added the explanation in Section 4.1.

Compared to INJ-CLIM, INJ-RF yields higher correlation coefficients and smaller RMSEs at most receptors, indicating that INJ-RF better captures the daily variability and peak values of total $PM_{2.5}$ concentrations during the fire seasons. However, INJ-RF improves the NMBs only in Darwin, Sydney (Liverpool), Melbourne (Alphington), and Newcastle. In other receptors, the total $PM_{2.5}$ concentrations are more underestimated in the INJ-RF experiment than in INJ-CLIM, possibly due in part to the neglect in our model setup of downward mixing of smoke in remote regions.

**Referee comment 2**

The authors implemented many of the reviewers' suggestions, and I am really impressed with the improvements made to this manuscript.

We thank the reviewer for the helpful comments and questions. Please refer to the revised manuscript with tracked changes. We respond to each specific comment in details below. The referee comments are shown in red. Our replies are shown in black and modified text is shown in blue. The annotated page and line numbers refer to the revised copy of the manuscript.

I have a few questions after seeing the new Figure S1. There are relatively few MISR plume records in the Southeast box (maybe ~150). How many of these were in the training/testing dataset? The "Top End" has many more and is therefore what the RF model is really trained on. What are the statistics if the results are separated for the two different regions? The seasonality for the two regions differ, and I wonder if the drivers would differ if the model was trained separately for the two regions (although there is not a large enough sample size to actually do this effectively for the Southeast). This also creates a bit of a mismatch in that the surface measurement sites are primarily in the Southeast box. It would be good to mention these

limitations as it may also explain why the comparisons with the surface concentrations are not much better.

The reviewer makes a good point. We acknowledge that the RF model is better trained on in northern Australia as the MISR plume records are relatively adequate in this region. In southeastern region, the fire season is shorter and there are only 254 MISR plume records. The training data consist of 244 records, and the testing data consist of 10 records. In the northern region, the correlation between the predicted and observed injected emission fluxes above the PBL is 0.7, the NMB is 7.3%, and the RMSE is $1.6\times10^{-10}$ kg m$^{-2}$ s$^{-1}$.  In the southeast region the correlation is 0.97, the NMB is 12%, and the RMSE is $6.1\times10^{-10}$ kg m$^{-2}$ s$^{-1}$, showing nearly as good agreement with MISR observations as in the north. We have discussed the limitations of the MISR dataset in the discussion and conclusion section. We have also added a new paragraph to Section 5, in which we highlight two of the main limitation of the RF model.

**P10, Line 264-268, Section 2.2.3**

A shortcoming of our machine-learning approach is that the MISR dataset used for our study includes relatively few plumes in southeastern Australia compared to northern Australia (Figure S1). The fire season is shorter in this region, and there is much greater interannual variability in fire activity. As a consequence, we have available only 244 training records in the southeast and only 10 for testing there, compared to 1447 records for training and 152 records for testing in the north. We further discuss this limitation in Section 5.

**P22-23, Line 674-682, Section 5**

The machine learning approach (INJ-RF) has two main limitations. First, the relatively short fire season and interannual variability of fire activity in southeastern Australia means that fewer MISR records are currently available to train and test the INJ-RF model. Digitizing more smoke plume records from MISR, a laborious process, could enhance the training and testing of the INJ-RF model.  Future studies could then train the random forest models separately in the two regions – northern and southeastern Australia – and identify the drivers for each region.  Second, as noted above, the STILT model cannot capture downward mixing of smoke away from source

regions. Future studies could explore the impacts of long-range transport and downward mixing of fire emissions on surface smoke concentrations by applying the estimated injection fractions to 3-D chemical transport models.

**A couple other revisions:**

1. Line 35: Change to "also leads to better model agreement". I think this should just be softened as that isn't clear in all the metrics at any of the sites.

Changed as suggested. Thank you.

**P2, Line 35-37, Abstract**

Using the plume behavior predicted by the random forest method also leads to better model agreement with observed surface $PM_{2.5}$ in several key cities near the wildfire source regions, with smoke $PM_{2.5}$ accounting for 5% to 52% of total $PM_{2.5}$ during fire seasons from 2009 to 2020.

2. Line 53-55: Change to "Smoke $PM_{2.5}$ is harmful to human health and the ambient environment..." OC and BC are part of $PM_{2.5}$ and there's also a lot of HAPs in smoke that are more toxic but not well-measured and likely in lower abundance.

Done.

**P3, Line 53-54, Introduction**

Smoke $PM_{2.5}$ is harmful to human health and the ambient environment (Reid et al., 2016; Aguilera et al., 2021; Johnston et al., 2021).

3. Line 82: change to "utilized for a long-term study."

Done. Thank you for pointing this out.

**P4, Line 79-81, Introduction**

The plume heights retrieved from TROPOMI offer daily global coverage, but TROPOMI data are available only from 2018 onwards and so cannot be utilized for a long-term study.

4. Line 88-89: The sentence "Besides these two studies…" feel out of place here. I'd reword to make it flow better.

Done.

**P4, Line 87-89, Introduction**

Besides these two methods for estimating injection height, Yao et al. (2018) used a machine learning model (random forest) and CALIOP data to predict the minimum heights of forest fire smoke in Canada.

5. Line 90: change to "represent"

Done.

**P4, Line 89-91, Introduction**

These three datasets represent the vertical extent of smoke plumes with high-resolution single parameters that specified the top and bottom heights of plumes, as well as the mean height of maximum injection (MHMI).

6. Line 91: add more explanation of MHMI

We have now included the definition and description of MHMI in Supplement Information.

**P4, Line 89-91, Introduction**

These three datasets represent the vertical extent of smoke plumes with high-resolution single parameters that specified the top and bottom heights of plumes, as well as the mean height of maximum injection (MHMI). The definitions of these variables are described in appendix S1.

7. Line 114: Change to "Alternatively, some studies use"

Done.

**P5, Line 114-116, Introduction**

Alternatively, some studies use atmospheric chemistry models to explicitly simulate smoke $PM_{2.5}$ concentrations from open fires and their impacts on air quality and health in Australia (Rea et al., 2016; Nguyen et al., 2020, 2021; Graham et al., 2021).

8. Line 117: Change to "of smoke air quality but may focus only on…"

Done.

**P5, Line 116-118, Introduction**

These studies can provide more accurate spatiotemporal variability of smoke air quality but may focus only on short-term simulations due to computational expense.

9. Line 138: Can likely remove this sentence about MERRA because it isn't really essential here (it focuses on the fact that it uses aerosols, but the authors are only using it for PBL).

Removed it as you suggested. Thanks.

**P6, Line 136-139, Section 2.1**

Daily mean PBL heights across Australia are obtained from the Modern-Era Retrospective Analysis for Research and Applications Version 2 (MERRA-2, Gelaro et al., 2017) at a spatial resolution of 0.5° latitude × 0.625° longitude. This reanalysis is often used to drive chemical transport models such as GEOS-Chem (Bey et al. 2001; Keller et al., 2014; Kim et al., 2015).

10. Line 349. I'd add something to remind the readers of what the first method described in Section 2.1 entails.

Done.

**P13, Line 356-358, Section 3.2**

Figure 2a compares the plume injection fractions above the PBL ($f_{abovePBL}$) derived from the MISR plume records with those calculated using the climatological plume profiles with assimilated PBL data (first method described in Section 2.1).

11. Figure 6: Can the pie carts just be made a bit smaller? I find the size visually unappealing.

Changed it as you suggested.

[Figure]

Figure 6. Contributions of simulated smoke PM$_{2.5}$ concentrations from the INJ-RF experiment to the observed total PM$_{2.5}$ concentrations (numbers on the pie charts) at 12 receptors averaged over the fire seasons of their respective observation periods (Table 3). Names of the observation sites are given in parentheses. Red sectors represent smoke contributions, while dark yellow sectors signify the differences between observed total PM$_{2.5}$ and simulated smoke PM$_{2.5}$ concentrations – i.e., the non-fire PM$_{2.5}$. Small circles on map represent the locations of these receptors. Different colors (red, blue, and black) are used to distinguish adjacent receptors.

12. I think Table S2 should be in the main text.

We have moved Table S2 to main text as new Table 3. Thank you.

Table 3. Statistics for daily mean $PM_{2.5}$ concentrations simulated by CTL, INJ-CLIM, and INJ-RF experiments, compared to the ground-based observations at 12 receptors. The daily mean concentrations are calculated over each receptor's observing period. Shown are the temporal correlation coefficients $R$, NMBs, and RMSEs of daily total $PM_{2.5}$ concentrations compared to the surface measurements.

| Cities (site) | Observation periods (Locations) | $R$ [a] | | | NMB | | | RMSE (µg m⁻³) | | |
|---|---|---|---|---|---|---|---|---|---|---|
| | | CTL | INJ-CLIM | INJ-RF | CTL | INJ-CLIM | INJ-RF | CTL | INJ-CLIM | INJ-RF |
| Darwin [b] (Palmerston) | 2011-2020 (130.94°E, 12.50°S) | 0.76 | 0.76 | 0.76 | 17.1% | -17.8% | -2.1% | 5.9 | 3.9 | 4.0 |
| Gladstone [c] (South Gladstone) | 2009-2020 (151.27°E, 23.86°S) | 0.57 | 0.55 | 0.55 | -5.4% | -11.0% | -11.2% | 1.9 | 1.7 | 1.7 |
| Brisbane [c] (Springwood) | 2009-2020 (153.13°E, 27.61°S) | 0.24 | 0.17 | 0.40 | 13.2% | 2.2% | -4.8% | 2.2 | 1.9 | 1.2 |
| Newcastle [d] (Wallsend) | 2009-2020 (151.66°E, 32.89°S) | 0.53 | 0.48 | 0.52 | 16.1% | 2.3% | -6.0% | 6.6 | 3.9 | 2.4 |
| Sydney [d] (Liverpool) | 2009-2020 (150.90°E, 33.93°S) | 0.40 | 0.38 | 0.37 | 14.8% | 6.8% | 0.4% | 4.8 | 3.4 | 2.8 |
| Wollongong [d] (Wollongong) | 2009-2020 (150.88°E, 34.41°S) | 0.27 | 0.28 | 0.27 | -6.3% | -11.6% | -14.7% | 1.5 | 1.4 | 1.6 |
| Melbourne [e] (Footscray) | 2009-2020 (144.87°E, 37.80°S) | 0.25 | 0.25 | 0.25 | -9.5% | -10.1% | -14.6% | 4.6 | 4.6 | 3.6 |
| Melbourne [e] (Alphington) | 2009-2020 (145.03°E, 37.77°S) | 0.41 | 0.39 | 0.40 | 23.7% | 22.9% | 14.4% | 2.0 | 2.0 | 1.2 |
| Albury [d] (Albury) | 2017-2020 (146.93°E, 36.05°S) | 0.93 | 0.93 | 0.93 | -22.2% | -23.7% | -31.7% | 5.6 | 5.8 | 7.3 |
| Canberra [f] (Florey) | 2014-2020 (149.04°E, 35.22°S) | 0.67 | 0.63 | 0.68 | 19.3% | -8.6% | -16.0% | 22.5 | 17.7 | 15.3 |
| Sydney [d] (Prospect) | 2014-2020 (150.91°E, 33.79°S) | 0.72 | 0.69 | 0.71 | 7.5% | -1.2% | -7.2% | 2.2 | 1.5 | 1.4 |
| Newcastle [d] (Newcastle) | 2014-2020 (151.75°E, 32.93°S) | 0.59 | 0.50 | 0.55 | 28.9% | 9.2% | -1.6% | 6.9 | 4.0 | 2.7 |

[a] Temporal correlation coefficient between the observed and simulated annual mean total $PM_{2.5}$ concentrations during the fire seasons (April to December for Darwin and Gladstone; August to December for other cities).
[b] Observation data source: Northern Territory Environment Protection Authority (http://ntepa.webhop.net/NTEPA/Default.ltr.aspx)
[c] Queensland Government Open Data Portal (https://apps.des.qld.gov.au/air-quality/download/)
[d] New South Wales Department of Planning and Environment (https://www.dpie.nsw.gov.au/air-quality/air-quality-data-services/data-download-facility)
[e] Victoria Environment Protection Authority (https://www.epa.vic.gov.au/for-community/airwatch)

[f] Australian Capital Territory Government Open Data Portal (https://www.data.act.gov.au/Environment/Air-Quality-Monitoring-Data/94a5-zqnn